# Embryonic transcription factor expression in mice predicts medial amygdala neuronal identity and sex-specific responses to innate behavioral cues

Julieta E Lischinsky[1,2], Katie Sokolowski[2], Peijun Li[2], Shigeyuki Esumi[2,3], Yasmin Kamal[2], Meredith Goodrich[2], Livio Oboti[2], Timothy R Hammond[2], Meera Krishnamoorthy[2], Daniel Feldman[2], Molly Huntsman[4], Judy Liu[2], Joshua G Corbin[2]*

[1]Institute for Biomedical Sciences, The George Washington University, Washington DC, United States; [2]Center for Neuroscience Research, Children's National Medical Center, Washington, DC, United States; [3]Graduate School of Medical Sciences, Kumamoto-University, Kumamoto City, Japan; [4]Department of Pediatrics, University of Colorado School of Medicine, Aurora, Colorado

**Abstract** The medial subnucleus of the amygdala (MeA) plays a central role in processing sensory cues required for innate behaviors. However, whether there is a link between developmental programs and the emergence of inborn behaviors remains unknown. Our previous studies revealed that the telencephalic preoptic area (POA) embryonic niche is a novel source of MeA destined progenitors. Here, we show that the POA is comprised of distinct progenitor pools complementarily marked by the transcription factors Dbx1 and Foxp2. As determined by molecular and electrophysiological criteria this embryonic parcellation predicts postnatal MeA inhibitory neuronal subtype identity. We further find that *Dbx1*-derived and Foxp2+ cells in the MeA are differentially activated in response to innate behavioral cues in a sex-specific manner. Thus, developmental transcription factor expression is predictive of MeA neuronal identity and sex-specific neuronal responses, providing a potential developmental logic for how innate behaviors could be processed by different MeA neuronal subtypes.

*For correspondence: JCorbin@cnmcresearch.org

## Introduction

One of the major functions of the limbic system is to integrate conspecific and non-conspecific environmental cues with social and survival salience to generate appropriate behavioral responses (*Sokolowski and Corbin, 2012*; *Stowers et al., 2013*). The medial subnucleus of the amygdala (MeA) serves as a hub in this function, residing only two synapses away from sensory neurons in the vomeronasal organ (*Dulac and Wagner, 2006*; *Sokolowski and Corbin, 2012*). The MeA along with the bed nucleus of the stria terminalis (BNST) and multiple nuclei of the hypothalamus including the ventromedial hypothalamus, form a core limbic circuit largely dedicated to processing innate behaviors (*Dulac and Wagner, 2006*; *Gross and Canteras, 2012*; *Choi et al., 2005*). Classical studies investigating patterns of neuronal activation in response to behavioral or olfactory cues (*Kollack and Newman, 1992*; *Erskine, 1993*), lesioning studies (*Vochteloo and Koolhaas, 1987*; *Takahashi and Gladstone, 1988*; *Kondo, 1992*) and more recent optogenetic approaches (*Hong et al., 2014*) have revealed a central role for the MeA in the regulation of innate behaviors such as aggression, mating and predator avoidance.

**eLife digest** Within the brain, a set of interconnected structures called the limbic system is involved in emotion, motivation and memory. This system – and in particular a structure called the medial amygdala – also contributes to behavioral drives that help an animal to survive and reproduce. These include the drive to avoid predators, to defend territory, and to find a mate. Such behaviors are thought to be inborn or innate. This means that animals display them instinctively whenever specific triggers are present, without the need to learn them beforehand.

However, just as a computer must be programmed to perform specific tasks, these innate behavioral responses must also be programmed into the brain. Given that animals do not learn these behaviors, Lischinsky et al. reasoned that specific events during the development of the brain must provide the animal's brain with the necessary instructions. To test this idea, they studied how the development of the medial amygdala in mouse embryos may give rise to differences in innate mating behavior seen between male and female mice.

The medial amygdala contains many subtypes of neurons, which show different responses to sex hormones such as estrogen and androgen. Lischinsky et al. show that two sets of cells give rise to some of the different neurons of the adult medial amygdala. One set of these precursor cells makes a protein called Dbx1 and the other makes a protein called Foxp2. These two sets of precursors generate medial amygdala neurons with different arrays of sex hormone receptors in male and female mice. Moreover, while the two sets of medial amygdala neurons are activated during aggressive encounters, they show different patterns of activation in male and female animals during mating.

These findings suggest that the development of *Dbx1*-derived and Foxp2+ neurons in the medial amygdala helps program innate reproductive and aggressive behaviors into the brain. The new findings also provide insights into why these behaviors differ in male and female mice. The next challenge is to identify the inputs and outputs of these two distinct subpopulations of medial amygdala neurons. This should make it possible to work out exactly how these populations of cells control innate behaviors in male and female animals.

In addition to the processing of innate cues, the MeA is one of many known sexually dimorphic regions of the brain, with differences in numbers of neurons, glia, and synaptic organization between males and females (*Cooke and Woolley, 2005*; *Johnson et al., 2008*; *McCarthy and Arnold, 2011*). The critical role that the MeA plays in regulating sex-specific behaviors is reflected in the high expression levels of steroid hormone pathway proteins such as aromatase, estrogen receptor and androgen receptor (*Wu et al., 2009*; *Juntti et al., 2010*; *Unger et al., 2015*). MeA neuronal subpopulations expressing different combinations of these proteins have been shown to regulate aggression or mating behaviors in male and female mice (*Juntti et al., 2010*; *Hong et al., 2014*; *Unger et al., 2015*). Nonetheless, understanding how developmental programs are linked to behavioral processing in the MeA remains unknown.

As unlearned behaviors are largely inborn, we reasoned that there must be embryonic developmental programs that guide the formation of sub-circuitry dedicated for different innate behaviors. Previous studies of MeA development revealed that progenitors located at the telencephalic-diencephalic border are a major source of MeA neuronal populations (*Zhao et al., 2008*; *Hirata et al., 2009*; *Soma et al., 2009*; *García-Moreno et al., 2010*). Our previous work revealed that one of these progenitor populations is defined by the transient expression of the developmentally regulated transcription factor, *Dbx1*, which in turn generates a subclass of MeA putative inhibitory projection neurons (*Hirata et al., 2009*). However, the MeA is also comprised of diverse populations of local interneurons and both excitatory and inhibitory output neurons (*Bian, 2013*; *Keshavarzi et al., 2014*). This suggests the contribution of other progenitor subpopulations, perhaps also originating from the POA to MeA neuronal diversity, populations which may in turn play different roles in innate behavioral processing.

Here, we demonstrate that in addition to Dbx1 expression, a subset of MeA embryonic progenitors are complementarily marked by expression of Foxp2, a forkhead transcription factor implicated

in the development and function of neurons and required in the motor coordinating centers of the brain for the appropriate production of speech (*French and Fisher, 2014*). We find this embryonic parcellation interestingly persists into postnatal stages where *Dbx1*-derived and Foxp2+ MeA neurons are separate, non-overlapping inhibitory output neuronal subpopulations. Strikingly, both subpopulations are activated during specific innate behaviors in a sex-specific manner. Thus, our findings link developmental patterning to innate behavioral processing and further provide an embryonic developmental framework for how these behaviors may emerge.

## Results

### Dbx1 and Foxp2 expression segregates embryonic and postnatal MeA subpopulations

Our previous work along with the work of others revealed that the telencephalic-diencephalic border is a major source of neurons that will populate the MeA (*Zhao et al., 2008*; *Hirata et al., 2009*; *Soma et al., 2009*; *García-Moreno et al., 2010*). Our previous studies (*Hirata et al., 2009*) revealed that the preoptic area (POA), which lies on the telencephalic side of this border (*Flames et al., 2007*), is a source of *Dbx1*+ progenitors fated to generate a subpopulation of MeA inhibitory output neurons. Our previous studies further revealed that progenitors arising from ventral telencephalic Shh+ and Nkx2.1+ domains also contributed to diverse neuronal subpopulations of the MeA (*Carney et al., 2010*). Thus, while a molecular map of MeA embryonic niche diversity is beginning to emerge, the diversity of mature neurons derived from this niche and whether there is a link between embryonic identity, mature identity and function remains unknown. Moreover, as these previously identified subpopulations only generate a subset of MeA neurons, we reasoned that there must be other transcription factor marked progenitor populations within the telencephalic-diencephalic niche.

Here, in addition to *Dbx1*+ progenitors, we observed a progenitor population comprised of Foxp2+ cells, residing primarily in the putative subventricular zone (SVZ) of the POA (*Figure 1a–f,s*). Interestingly, during embryogenesis, *Dbx1*-derived and Foxp2+ progenitor populations were non-overlapping (*Figure 1a–l*). Both populations were also generally distinct from OTP+ progenitors (*Figure 1—figure supplement 1a–i*), a population previously shown to define a subset of MeA-fated progenitors (*García-Moreno et al., 2010*). We next investigated whether Foxp2+ progenitors overlapped with ventral telencephalic populations derived from *Shh* or *Nkx6.2* lineages, which also encompass the POA (*Carney et al., 2010*; *Fogarty et al., 2007*). We found embryonic Foxp2+ cells were not derived from either lineage (~5% overlap) (*Figure 1—figure supplement 2*) further expanding our knowledge of the molecular diversity of the MeA niche.

Interestingly, this embryonic molecular parcellation persisted into adulthood as *Dbx1*-derived and Foxp2+ cells remained non-overlapping across the rostro-caudal extent of the postnatal MeA (*Figure 1m–r*; *Figure 1—figure supplement 3*). Similarly, postnatally, the *Dbx1*-derived and Foxp2+ neurons remained distinct from OTP+ cells (*Figure 1—figure supplement 1j–x*). Taken together, these findings reveal that *Dbx1*-derived and Foxp2+ populations, although appearing to derive from the same embryonic niche, remain distinct subpopulations from embryonic development to adulthood (*Figure 1t*), a novel finding that we hypothesize has implications for later subtype identity and function, explored in the next sets of experiments.

### Foxp2+ neurons in the postnatal amygdala are inhibitory

Our previous work revealed that MeA *Dbx1*-derived neurons are a subclass of inhibitory neurons, likely projection as opposed to local interneurons (*Hirata et al., 2009*). However, the identity of MeA Foxp2+ neurons remains unknown. Therefore, we next examined whether adult MeA Foxp2+ cells were neurons or glia. Our analysis revealed that 81% ± 2.6 of Foxp2+ cells expressed NeuN, a pan neuronal marker (*Mullen et al., 1992*) (*Figure 2a–d*), with none co-expressing the oligodendrocyte marker, CC1 (*Koenning et al., 2012*) (*Figure 2e–h*). We next wanted to determine if Foxp2+ neurons were excitatory or inhibitory. We found that only 3% ± 1.2 of Foxp2+ cells were derived from the *Emx1*-lineage, a broad marker of excitatory neurons (*Gorski et al., 2002*) (*Figure 2i–l*). We further found that 22% ± 6.6 of Foxp2+ cells expressed the inhibitory marker Calbindin (*Figure 2m–p*), with a smaller percentage of Foxp2+ cells (15% ± 2.5) expressing nNOS (*Figure 2q–t*), or

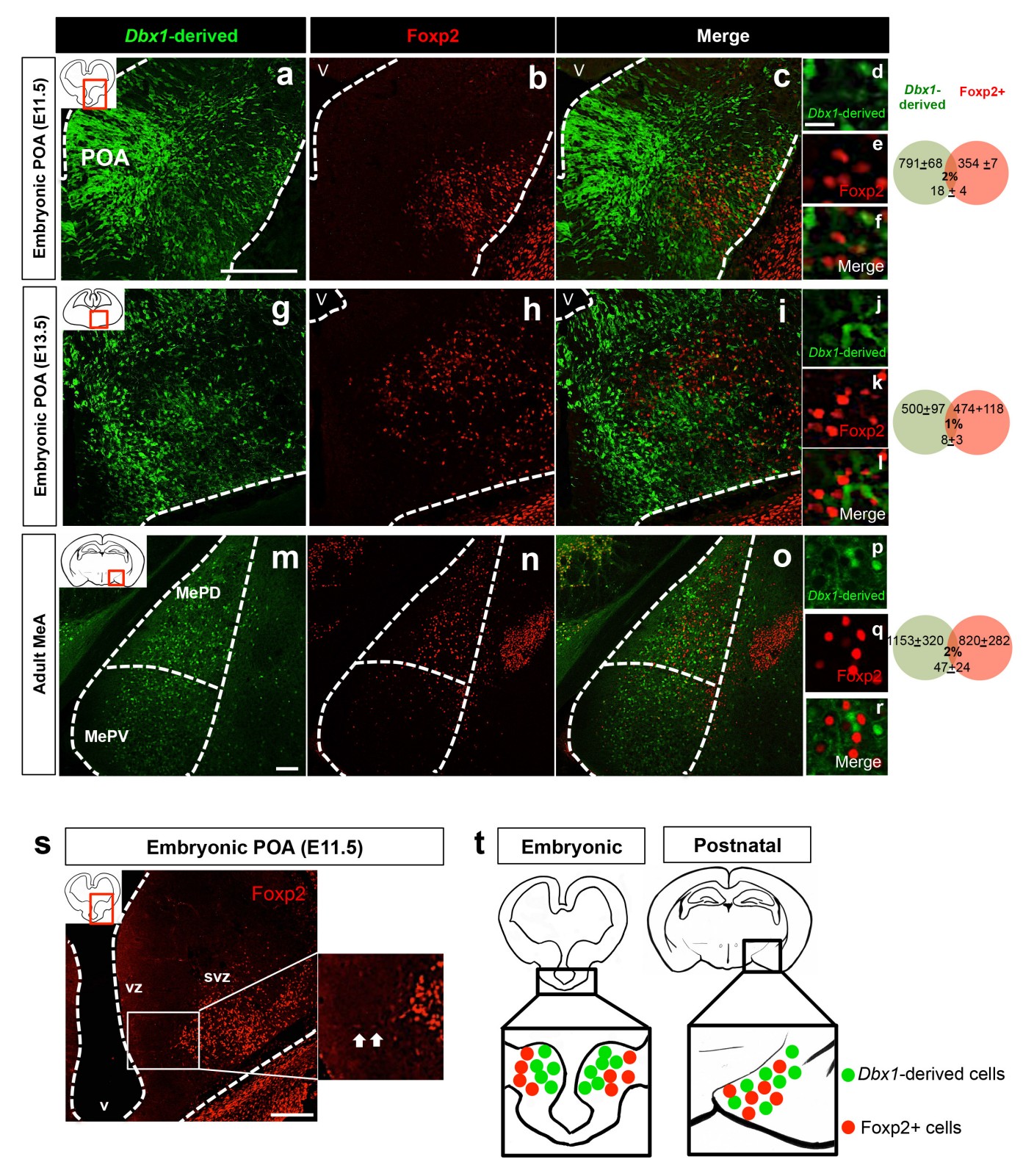

**Figure 1.** Embryonic and postnatal segregation of *Dbx1*-derived and Foxp2+ cells. Minimal co-localization of *Dbx1*-derived (green) and Foxp2+ (red) labeled cells in coronal sections at the level of the POA at E11.5 (**a–f**) and E13.5 (**g–l**). As shown at E11.5 (**s**) Foxp2+ precursors in the VZ (arrows) are observed putatively migrating to the SVZ. The segregation of *Dbx1*-derived and Foxp2+ cells persists into adulthood (**m–r**). Summary schematic of spatial segregation of *Dbx1*-derived and Foxp2+ cells in the embryonic POA and postnatal MeA (**t**). Venn diagram depicting the overlap of the total

*Figure 1 continued on next page*

*Figure 1 continued*

number of *Dbx1*-derived and Foxp2+ cells in 2–3 sections/embryo or 5–8 sections/adult brain. The scale bars represent 200 μm (**a–c, g–i, s**), 100 μm (**m–o**) and 25 μm (**d–f, j–l, p–r**). Abbreviations: MeA, medial amygdala; MePD, medial amygdala-posterior dorsal; MePV, medial amygdala posterior ventral; POA, preoptic area; SVZ, subventricular zone; V, ventricle; VZ, ventricular zone. *n* = 3 embryonic brains; *n* = 5 postnatal brains. Data are mean ± s.e.m. *n* is the number of animals.

The following figure supplements are available for figure 1:

**Figure supplement 1.** Embryonic and postnatal distribution of OTP+ cells.

**Figure supplement 2.** Foxp2+ cells are not derived from the *Shh*- or *Nkx6.2* lineages.

**Figure supplement 3.** Localization of *Dbx1*-derived and Foxp2+ cells in the adult MeA.

somatostatin (5% ± 0.6) (*Figure 2u–x*), inhibitory markers that mark a subset of MeA output neurons (*Tanaka et al., 1997*) and interneurons (*Ascoli et al., 2008*), respectively. Collectively, although we did not fully assess all putative inhibitory markers, these data reveal that Foxp2+ MeA are not excitatory and are likely inhibitory (*Figure 2y*).

As neither *Dbx1*-derived nor Foxp2+ cells expressed OTP+, we therefore investigated the identity of this population. We found that 1% ± 0.4 of OTP+ cells were derived from the *Emx1*-lineage, and 13.5% ± 9.1 co-expressed CAMKIIα (*Jones et al., 1994*), both excitatory markers. In contrast, we observed 56% ± 18.9 of OTP+ cells co-expressed calbindin, while none (0% ± 0.1) co-expressed somatostatin (*Figure 2—figure supplement 1*). Therefore, similar to *Dbx1*-derived and Foxp2+ neurons, the majority of OTP+ cells appear to be inhibitory.

## *Dbx1*-derived and Foxp2+ neurons possess distinct electrophysiological properties

To determine if the *Dbx1*-derived and the Foxp2+ populations are functionally distinct subclasses, we next examined their electrophysiological properties. Previous studies (*Bian, 2013*; *Keshavarzi et al., 2014*) revealed a significant diversity in intrinsic electrophysiological properties of MeA local and projection neurons. Here, we found that the majority (19/28) of *Dbx1*-derived neurons were characterized by a regular, tonic spiking pattern with 3–4 spikes at rheobase (*Figure 3a*). In contrast, the majority (15/23) of Foxp2+ neurons were distinguished by a phasic firing pattern and displayed a single or double spike upon repolarization after hyperpolarization, a profile characteristic of inhibitory neurons (*Llinás, 1988*) (*Figure 3b*). *Dbx1*-derived and *Foxp2*-derived neurons (confirmed Foxp2+ by immunohistochemistry) also displayed significant differences in resting membrane potential, input resistance, capacitance, and action potential frequency but not in rheobase (*Figure 3c–g*). In addition, the presence of spines in Foxp2+ neurons (*Figure 3—figure supplement 1*) suggested that similar to *Dbx1*-derived neurons, Foxp2+ neurons are projection neurons. This reveals that despite both populations being inhibitory, the *Dbx1*-derived and the Foxp2+ populations possess distinct firing patterns.

We further analyzed spontaneous excitatory post-synaptic currents (sEPSCs), a measure of excitatory inputs. *Dbx1*-derived neurons received significantly more frequent and greater amplitude of sEPSCs than Foxp2+ neurons (*Figure 3h–l*). This suggests that *Dbx1*-derived MeA neurons receive a greater number and/or stronger excitatory inputs than Foxp2+ neurons. In summary, a combination of neuronal marker expression (*Figure 2*) and electrophysiological (*Figure 3*) analyses, combined with our previous analysis (*Hirata et al., 2009*) revealed that *Dbx1*-derived and Foxp2+ neurons are distinct subclasses of inhibitory, and are likely projection neurons.

## Molecular identity of *Dbx1*-derived and Foxp2+ postnatal MeA cells

Based on the above analyses revealing that *Dbx1*-derived and Foxp2+ neurons are separate subclasses, we next wanted to determine whether these two populations express different combinations of steroid pathway proteins previously associated with MeA function such as estrogen receptor-alpha (ERα), aromatase and androgen receptor (AR) (*Wu et al., 2009*; *Juntti et al., 2010*; *Unger et al., 2015*). As the MeA is a sexually dimorphic nucleus (*Cooke and Woolley, 2005*;

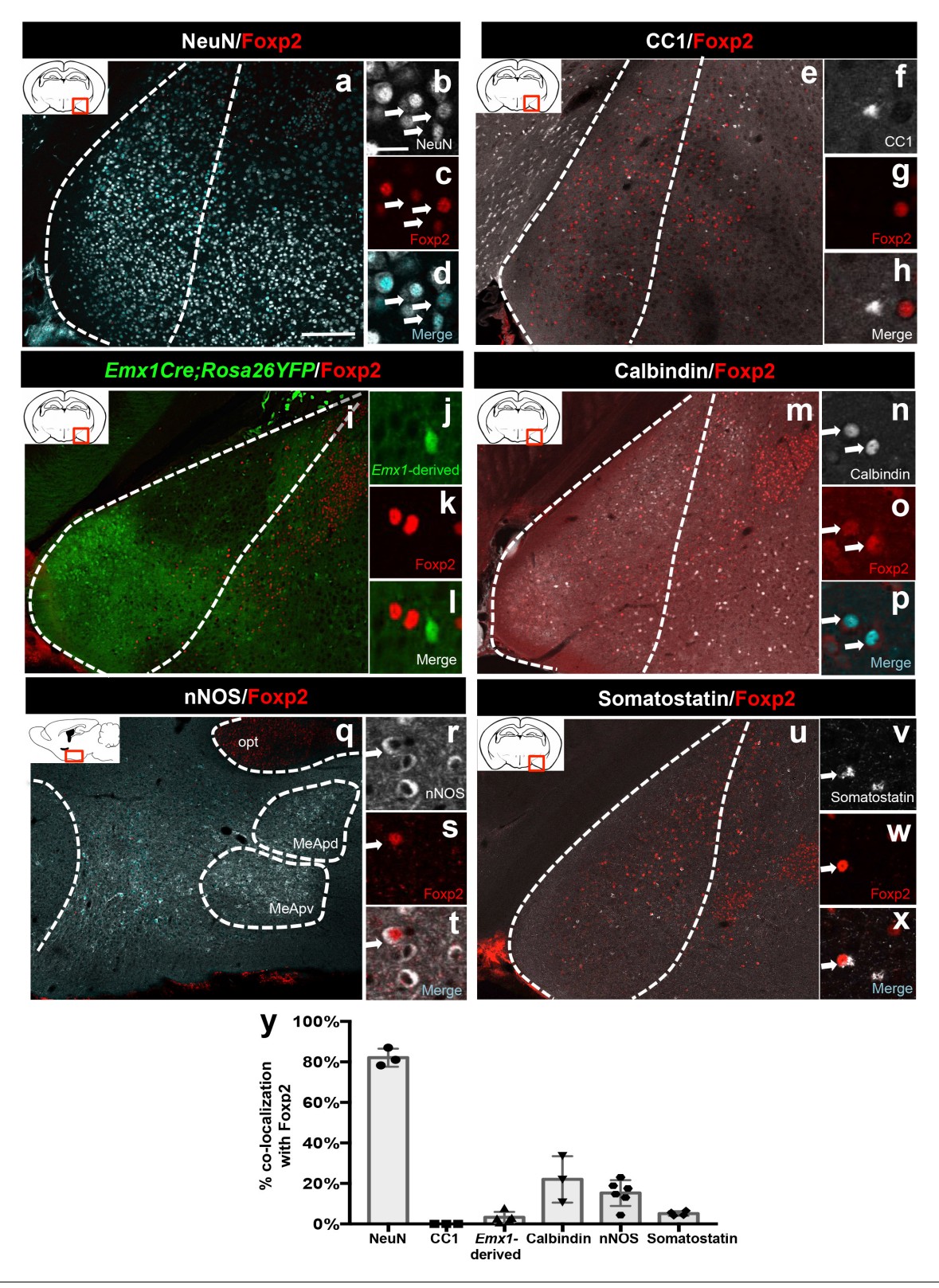

**Figure 2.** Identity of Foxp2+ medial amygdala neurons. Dual immunofluorescence of Foxp2 with NeuN (*n* = 3 mice) (**a–d**), CC1 (*n* = 3 mice) (**e–h**), YFP (in *Emx1^cre^;RYFP* mice) (*n* = 5 mice) (**i–l**), calbindin (*n* = 3 mice) (**m–p**), nNOS (*n* = 6 mice) (**q–t**), and somatostatin (*n* = 4 mice) (**u–x**) (arrows show double-labeled cells). Quantification of co-localization with each marker (**y**). Data are mean ± s.e.m. *n* is the number of animals. The scale bar represents 200 μm (**a, e, i, m, q, u**) and 25 μm (**b–d, f–h, j–l, n–p, r–t, v–x**).

*Figure 2 continued on next page*

*Figure 2 continued*

The following figure supplement is available for figure 2:

**Figure supplement 1.** Identity of OTP+ medial amygdala neurons.

*McCarthy and Arnold, 2011*; *Johnson et al., 2008*), we characterized the expression of these markers in both male and female mice (*Figure 4*, *Figure 4—figure supplement 1*, *Figure 4—figure supplement 2*). We found that *Dbx1*-derived and Foxp2+ cells in males expressed ERα to the same extent (28.4% ± 4.8 in *Dbx1*-derived cells; 24.0% ± 7.2 in Foxp2+ cells). However, *Dbx1*-derived and Foxp2+ cells in females showed significant differences in ERα expression (45% ± 3.4 in *Dbx1*-derived cells; 24.8% ± 5.8 in Foxp2+ cells) (*Figure 4a–g*). The majority of *Dbx1*-derived cells expressed aromatase both in males (61.7% ± 7.6) and females (52.4% ± 5.2), which was at a significantly higher percentage than in Foxp2+ cells in males (0.12% ± 0) and females (7.2% ± 6.0) (*Figure 4h–n*). There were no subpopulation differences in AR expression as both *Dbx1*-derived and Foxp2+ neurons in both males (26.8% ± 4.1 in *Dbx1*-derived cells; 16.2% ± 3.2 in Foxp2+ cells) and females (8.4% ± 3.6 in *Dbx1*-derived cells; 7.0% ± 1.8 in Foxp2+ cells) co-expressed AR at the same levels (*Figure 4o–u*). However, there were sex-specific differences observed as a greater percentage of *Dbx1*-derived cells in males (26.8% ± 4.1) expressed AR than *Dbx1*-derived cells in females (8.4% ± 3.6).

We also analyzed the contribution of the *Dbx1*-derived and Foxp2+ populations to the total ERα, Aromatase and AR MeA populations (*Figure 4—figure supplement 2*). We observed that both *Dbx1*-derived and Foxp2+ cells comprised between ~10% to 22% of the total ERα population in male and female mice. The *Dbx1*-derived population contributed between ~30–40% of the total aromatase population in males and females, which was significantly greater than the contribution of the Foxp2+ population in both males and females. *Dbx1*-derived (males only) and Foxp2+ cells (males and females) comprised ~20–38% of the total AR+ population. In contrast, the *Dbx1*-derived population contributed to only 3% of the total AR+ population in females, which was significantly less than the Foxp2+ contribution in females and less than the contribution of *Dbx1*-derived cells in males. Collectively, these data reveal that *Dbx1*-derived and Foxp2+ cells contributed differentially to the aromatase and AR populations, but not to the ERα population.

## Sex-specific subtype activation patterns during innate behaviors

The MeA receives direct inputs from the accessory olfactory bulb (AOB) (*Scalia and Winans, 1975*; *Martel and Baum, 2009*; *Bergan et al., 2014*) and integrates this chemosensory information to regulate innate behaviors including territorial aggression, maternal aggression, mating, and predator avoidance (*Dulac and Wagner, 2006*; *Kim et al., 2015*). Previous data revealed that at least aggressive and mating behaviors are controlled by MeA GABAergic neurons (*Choi et al., 2005*; *Hong et al., 2014*). However, whether different subsets of inhibitory neurons are activated during these behaviors, or if neuronal subtype activation is generalizable across behaviors remains unknown. To directly test these possibilities we performed well characterized aggression, mating and predator odor avoidance behavioral tests in both male and female resident mice and examined the patterns of activation of *Dbx1*-derived and Foxp2+ cells using c-fos as a readout of neuronal activity in resident mice (*Figure 5—figure supplement 1*).

### *Dbx1*-derived and Foxp2+ neurons are activated during aggressive encounters

To evaluate activation of *Dbx1*-derived and Foxp2+ neurons during male conspecific aggression, we performed a territorial aggression assay in which an intruder mouse was placed in a resident cage. Concordant with previous literature (*Wang et al., 2013*), we found a significant increase in the number of c-fos+ cells in the MeA in comparison to naïve mice (*Figure 5a–c*). In addition, both the number and proportion of activated *Dbx1*-derived and Foxp2+ subpopulations in males were significantly increased during territorial aggression compared to control (naïve) mice (*Figure 5d–k, x*). We next evaluated aggression in female mice by conducting a maternal aggression assay in which

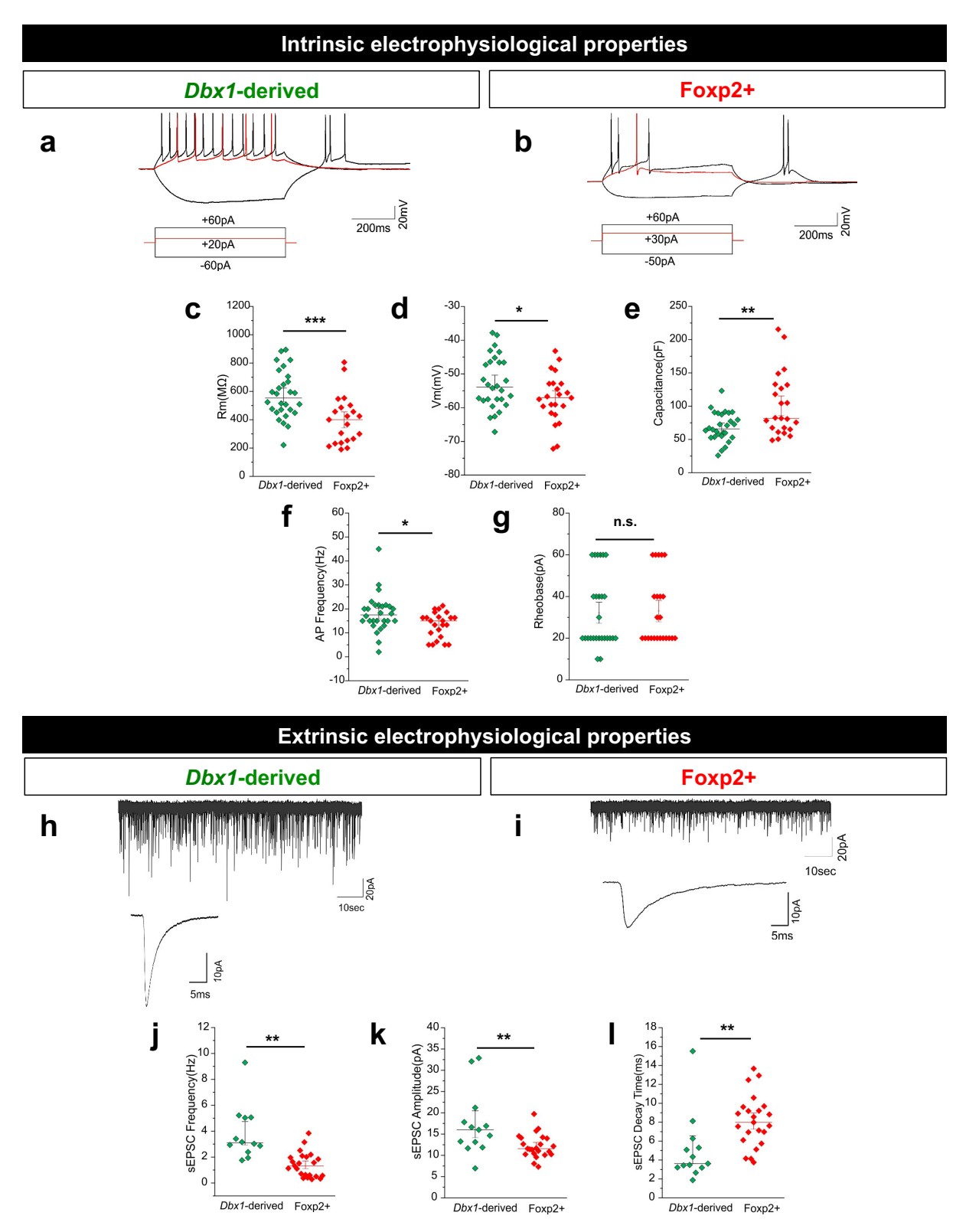

**Figure 3.** *Dbx1*-derived and Foxp2+ MeA neurons possess distinct electrophysiological properties. Typical firing patterns of *Dbx1*-derived (a) and Foxp2+ (b) neurons with current injections at −60 pA,+20 pA and +60 pA. Significant differences across populations in resting membrane potential (Rm) (c), voltage at rest (Vrest) (d), capacitance (e), and action potential (AP) firing patterns (f) but not rheobase (g) are observed (two-tailed *t*-test, (c) p=0.0006, t = 3.676, df = 49; (d) p=0.036, t = 2.159, df = 49; (e) p=0.002, t = 3.1987, df = 49; (f) p=0.016, t = 3.1988, df = 49; (g) p=0.610, t = 0.514,

*Figure 3 continued on next page*

*Figure 3 continued*

df = 49; *n* = 28 *Dbx1*-derived cells and *n* = 23 Foxp2+ cells). Spontaneous excitatory post-synaptic currents (sEPSCs) are observed in both *Dbx1*-derived (**h**) and Foxp2+ (**i**) neurons, with significant differences in sEPSC frequency (**j**), amplitude (**k**) and decay (**l**) (two-tailed *t*-test, (**j**) p=<0.0001, t = 3.041, df = 34; (**k**) p=0.006, t = 2.949, df = 34; (**l**) p=0.005, t = 3.041, df = 34; *n* = 28 *Dbx1*-derived neurons, *n* = 23 Foxp2+ neurons). Data are mean ± s.e.m. *n* is the number of cells. p<0.05 (*), p<0.01 (**), and p<0.001 (***), n.s.; not significant.

The following figure supplement is available for figure 3:

**Figure supplement 1.** Foxp2+ neurons possess projection neuronal morphology.

pups were removed from a nursing female and a sexually naïve male intruder was introduced (*Haney et al., 1989*). In addition to the naïve control, we established a second control in which pups were removed and no intruder was presented. With this control, we could compare levels of neuronal activation during maternal aggression (pups removed and presence of intruder) to levels of neuronal activation in response to a strong stressor (pups removed but no intruder) and to a naïve condition (with pups and without intruder). We found a significant increase in the number of c-fos+ cells in the MeA in the maternal aggression condition compared to both stressed and naïve controls (*Figure 5l–o*). When examining subtype-specific levels of activation, we found that both the number and proportion of activated *Dbx1*-derived and the Foxp2+ subpopulations significantly increased during maternal aggression in comparison to the naïve condition (*Figure 5p–w,y*). Thus, during an aggressive encounter with a conspecific, both *Dbx1*-derived and Foxp2+ MeA subpopulations were activated to a greater extent in both male and female mice.

## *Dbx1*-derived and Foxp2+ neurons are activated in a sex-specific manner during mating

We next conducted male and female mating assays and monitored the animals for mating (mounting and intromission followed by the presence of a vaginal plug). Consistent with previous studies (*Rowe and Erskine, 1993*), we observed an increase in c-fos+ cells in the MeA during both male and female mating (*Figure 6a–c*). Intriguingly, we found subpopulation specific differences in activation patterns across sexes. While the number and proportion of the activated *Dbx1*-derived subpopulation was significantly increased during both male and female mating when compared to naïve controls (*Figure 6d–g,l*), the number and proportion of activated Foxp2+ cells increased only during male but not during female mating (*Figure 6h–k,m*). Thus, while *Dbx1*-derived and Foxp2+ MeA subpopulations were both activated during male mating, only *Dbx1*-derived neurons were activated during female mating.

## Activation of *Dbx1*-derived and Foxp2+ neurons during predator odor exposure

We next sought to determine whether *Dbx1*-derived and Foxp2+ MeA subpopulations were activated by a strong innate stressor. To accomplish this, we exposed mice to rat odor, a well-characterized predator cue that evokes a strong aversive response in mice (*Apfelbach et al., 2005*; *Carvalho et al., 2015*; *Sokolowski et al., 2015*). Mice were exposed to soiled bedding from a rat cage (predator) or clean bedding (benign) as a control (*Figure 7*). Consistent with previous studies (*Canteras et al., 2015*), in response to predator odor we observed a significant increase in the number of c-fos+ cells in the MeA in both male and female mice (*Figure 7a–c*). Interestingly, the *Dbx1*-derived cells in male and female mice were not activated in response to predator odor exposure in comparison to the benign bedding (*Figure 7d–g*). In contrast, we observed a significant activation of Foxp2+ cells in comparison to controls in female mice, but not in males (*Figure 7h–k*). When we assessed the percentage of both *Dbx1*-derived (YFP+;c-fos+/total YFP+) and Foxp2+ (Foxp2+;c-fos+/total Foxp2+) cells co-labeled with c-fos over the total population there was no significant difference in either subpopulation nor in male or female mice compared to control (*Figure 7m–l*). This suggests that despite an increase in the number of activated Foxp2+ cells in female mice in the presence of rat bedding, this increase might not be physiologically relevant. Therefore, to investigate which MeA subpopulation may be responding to predator odor, we next examined the activation

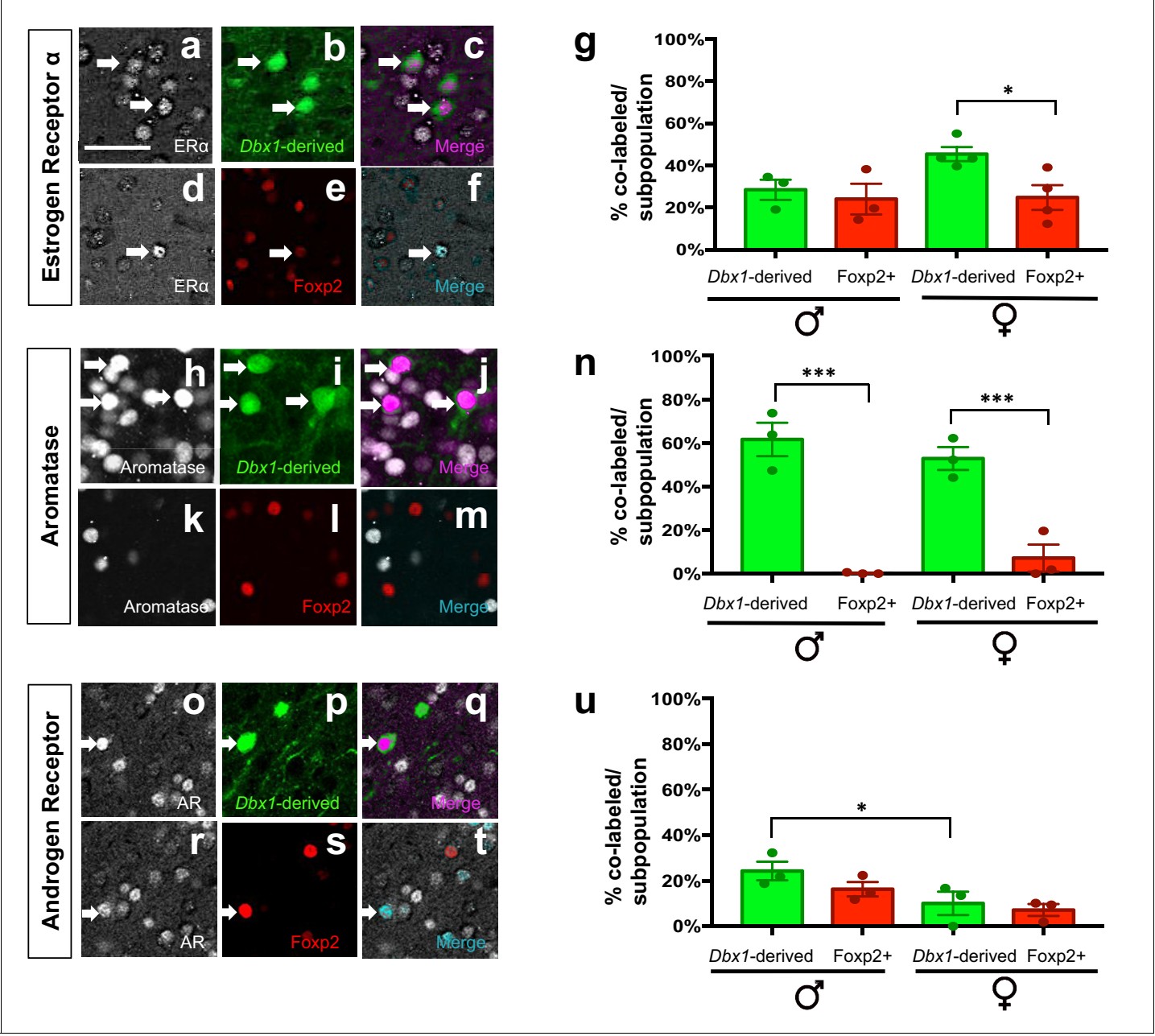

**Figure 4.** Expression of sex hormone pathway markers in *Dbx1*-derived and Foxp2+ cells. Dual immunofluorescence for YFP (green) or Foxp2 (red) with the sex steroid hormone pathway markers (white): estrogen receptor α (ERα) (**a–g**), aromatase (**h–n**) or androgen receptor (AR) (**o–u**) (arrows show double-labeled cells). Subpopulations of both *Dbx1*-derived (**a–c, g**) and Foxp2+ cells express ERα (**d–g**). Quantification reveals a greater percentage of *Dbx1*-derived cells expressing ERα compared to Foxp2+ cells in females (two-way ANOVA with no interaction between subpopulation and sex, but with main effect for subpopulation p=0.0433, F(1,10)=5.352; n = 4 and x = 45.38 for *Dbx1*-derived cells in female mice, n = 4 and x = 24.77 for Foxp2+ cells in female mice, n = 3 and x = 28.43 for *Dbx1*-derived cells in male mice, n = 3 and x = 24 for Foxp2+ cells in male mice). The majority of *Dbx1*-derived cells express aromatase in both males and females (**h–j, n**). In contrast, only a small percentage of Foxp2+ cells in both males and females express aromatase (two quantile regression analysis for non-normal distributions shows no interaction for sex and subpopulation but a main effect for subpopulation p=0.000 with a 95% confidence interval of 31.27 and 69.53, n = 3 for *Dbx1*-derived cells in male mice, n = 3 and for Foxp2 + cells in male mice, n = 3 for mice for *Dbx1*-derived cells in female mice, n = 3 and for Foxp2+ female mice). A greater percentage of *Dbx1*-derived neurons in males express AR in comparison to *Dbx1*-derived cells in female mice (**o–q, u**). No differences in percentages of Foxp2+ male or female subpopulations expressing AR (**r–u**) nor differences across subpopulations were observed (two-way ANOVA with no interaction between subpopulation and sex, but with main effect for sex p=0.0166, F(1,8)=9.118; n = 3 and x = 10.07 for *Dbx1*-derived cells in female mice, n = 3 and x = 7.133 for Foxp2+ cells in female mice, n = 3 and x = 24.3 for *Dbx1*-derived cells in male mice, n = 3 and x = 16.27 for Foxp2+ cells in male mice). Data are mean ± s.e.m. n is the number of animals and x is the mean. The scale bar represents 50 μm. p<0.05 (*) and p<0.01 (**).

*Figure 4 continued on next page*

*Figure 4 continued*

The following figure supplements are available for figure 4:

**Figure supplement 1.** Patterns of MeA expression of sex steroid hormone markers.

**Figure supplement 2.** Percent contribution of *Dbx1*-derived and Foxp2+ cells to sex steroid hormone marker populations.

patterns of the OTP+ population. We observed that a greater number of OTP+ cells in both males and females were activated in response to predator odor (*Figure 7—figure supplement 1*). When analyzing the proportion of OTP+ cells activated, we found no differences between the benign and the predator avoidance groups in males, but we did find a significant increase in the OTP+ population in females. Therefore, in the female brain, the OTP+ population, in contrast to *Dbx1*-derived or Foxp2+ cells, were activated at a level above control in response to a strong aversive innate olfactory cue.

In summary, our analyses of activation patterns in response to a battery of innate behavior tasks revealed that *Dbx1*-derived and Foxp2+ cells in the MeA were differentially activated depending on the instinctive behavioral task (*Figure 8*). The most striking of these differences occurred during mating behaviors. While both subpopulations were activated during male mating, in the female MeA only *Dbx1*-derived neurons were activated, with no activation of the Foxp2+ population. In contrast, the OTP+ population in females appears more tuned to a predator odor cue. Importantly, sex differences in patterns of neuronal activation during mating and predator odor were not due to overall differences in the activation of the MeA as these cues activated the MeA in both sexes. Taken together with our electrophysiological and molecular findings of the *Dbx1*-derived and Foxp2+ populations, our data reveal that the developmental parcellation of MeA progenitors predicts mature neuronal identity and sex-specific innate behavioral activation patterns.

## Discussion

Across a variety of species, innate behaviors such as aggression, mating and avoidance of predators are initiated by sensory cues primarily detected by olfaction (*Dulac and Wagner, 2006*; *Stowers et al., 2013*). Here, focusing on the medial amygdala (MeA), a critical brain region for the processing of olfactory-based sensory cues for unlearned behavior in vertebrates, we took a neural developmental approach to shed light on how innate behavioral information may be encoded in the male and female brain. Integrating genetic fate-mapping, patch clamp electrophysiology and animal behavioral assays we uncover a fundamental link between embryonic patterning and brain responses to innate behavioral cues at two levels: (1) differential transcription factor expression within the embryonic MeA progenitor niche predicts mature output neuronal subtype identity and (2) further predicts subpopulation sex-specific responses to mating and predator odor avoidance cues. Our findings further suggest that transcription factor expression at the progenitor stage may be instructive for the establishment of neuronal populations and sub-circuits regulating sex-specific behaviors.

### Potential transcription factor codes for establishment of MeA neuronal diversity

The generation of neuronal diversity across amygdala subnuclei has been posited to occur in a compartmentalized manner with amygdala inhibitory neurons generated in the subpallial ganglionic eminences and excitatory neurons arising from the cortical pallial region (*Swanson and Petrovich, 1998*). In this model, the amygdala and cerebral cortex develop by a similar mechanism with neurons in both structures originating in shared progenitor domains. However, more recent studies have revealed that the generation of amygdala neuronal diversity is more complex with large populations of neurons originating in progenitor niches dedicated for limbic structures (*Remedios et al., 2007*; *Hirata et al., 2009*; *Soma et al., 2009*; *Waclaw et al., 2010*; *García-Moreno et al., 2010*). One of these major niches encompasses the region at the telencephalic-diencephalic border, an origin of MeA output neurons (*Hirata et al., 2009*; *García-Moreno et al., 2010*). Our previous studies

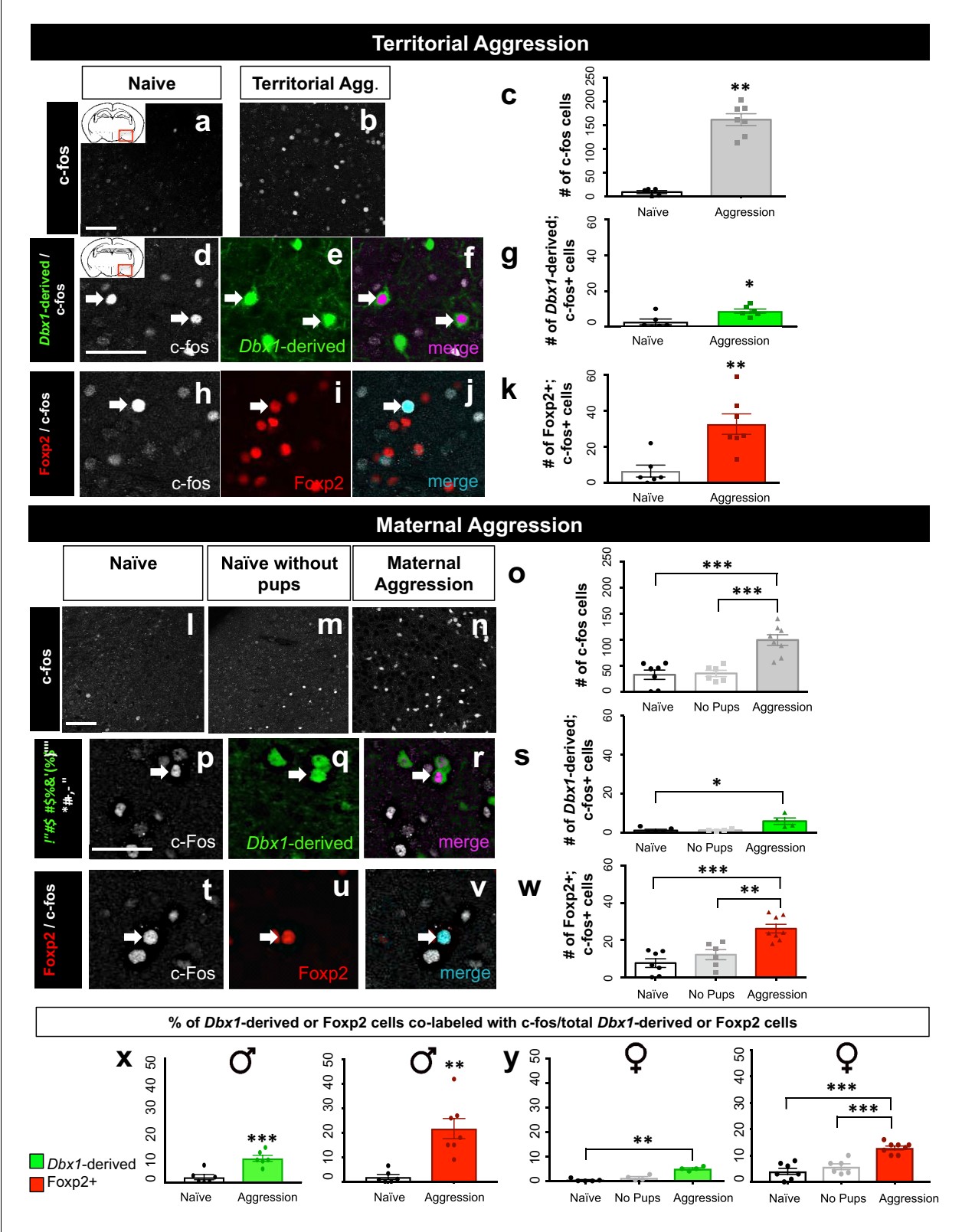

**Figure 5.** *Dbx1*-derived and Foxp2+ neurons are activated during aggressive encounters. Significant increases in the number of c-fos+ cells (white) compared to controls are observed in the MeA during both male territorial aggression (**a–c**) (two tailed Mann-Whitney test, U = 0, p=0.0025, *n* = 5 naïve mice, *n* = 7 territorial aggression) and maternal aggression (one-way ANOVA, p=<0.0001, F(2, 18)=19.05, *n* = 7 naïve mice, *n* = 6 'no pup' condition, *n* = 8 maternal aggression) (**l–o**). Dual immunofluorescence for c-fos (white) and YFP (green) or Foxp2 (red) after male territorial aggression

*Figure 5 continued on next page*

*Figure 5 continued*

reveals an increase in the number of *Dbx1*-derived cells (two-tailed Mann-Whitney test, U = 4, p=0.0238, *n* = 6) (**d–g**) and Foxp2+ cells (two-tailed Mann-Whitney test, U = 0, p=0.0023 *n* = 7) expressing c-fos in comparison to naïve control mice (*n* = 6) not exposed to an intruder (**h–k**). In female mice, an increase in the number *Dbx1*-derived cells expressing c-fos during maternal aggression (*n* = 4) in comparison to naïve (*n* = 5), but not in the 'no pups' conditions (*n* = 4), is observed (Kruskal-Wallis test, $X^2$(3,13)=7.124 p=0.0164) (**p–s**). There is also an increase in the number of Foxp2+ cells (*n* = 8) expressing c-fos during maternal aggression in comparison to the naïve (*n* = 7) and 'no pups' control conditions (*n* = 6) (one-way ANOVA p=<0.0001, F(2,18)=16.93) (**t–w**). Analyses of the percentage of both *Dbx1*-derived (YFP+ and c-fos+ /total YFP+) and Foxp2+ cells (Foxp2+ and c-fos+/ total Foxp2+) expressing c-fos in male mice reveals an increase in both the *Dbx1*-derived subpopulation (*n* = 6) (two-tailed *t*-test, p=0.0009, t = 4.685, df = 11) and the Foxp2+ subpopulation (*n* = 7) (two-tailed *t*-test, p=0.001, t = 4.295, df = 11) in comparison to naïve controls (*n* = 6) (**x**). In females, the percentage of *Dbx1*-derived cells expressing c-fos during maternal aggression is higher in comparison to the naïve condition but not to the 'no pups' condition (Kruskal-Wallis test, $X^2$(3,13)=9.461 p=0.0005, *n* = 5 naïve, *n* = 4 'no pups' and *n* = 4 aggression) (**y**). The percentage of Foxp2+ cells expressing c-fos during maternal aggression is also higher during maternal aggression (*n* = 8) in comparison to both the naïve (*n* = 7) and 'no pups' (*n* = 6) controls (one way ANOVA, p=<0.0001, F(2,18)=24.16) (**y**). Data are mean ± s.e.m. *n* is the number of animals. Arrows show double-labeled cells. p<0.05 (*), p<0.01 (**), and p<0.001 (***). The scale bar represents 50 μm.

The following figure supplement is available for figure 5:

**Figure supplement 1.** c-fos expression in the posterio-medial (MePD) and posterio-ventral (MePV) subnuclei during different innate behavioral assays.

revealed that the homedomain encoding transcription factor, *Dbx1*, marks a subpopulation of progenitors within the POA, which will later generate a subset of MeA inhibitory output neurons (*Hirata et al., 2009*). Here, we significantly extend this work by revealing the presence of a complementary population of progenitors within this niche marked by expression of Foxp2. Thus, our findings, combined with previous work, suggest a model of MeA development in which distinct progenitor populations at the telencephalic-diencephalic border defined by differential transcription factor expression (e.gs. Dbx1, Foxp2, OTP) are a major source for MeA neuronal diversity.

The function of combinatorial sets of transcription factors in neural progenitors has been shown across the neuraxis as the mechanism for the generation and specification of distinct subclasses of neurons (*Kepecs and Fishell, 2014*; *Stepien et al., 2010*; *Shirasaki and Pfaff, 2002*). In addition to specification of neuronal subtype identity, recent studies in the spinal cord and globus pallidus (*Dodson et al., 2015*; *Bikoff et al., 2016*) have revealed that different combinatorial codes in progenitor pools predict neuronal subtype connectivity patterns, neuronal firing properties and in the case of the globus pallidus, distinct functions in regulating voluntary movements (*Dodson et al., 2015*). Thus, transcription factor expression at the earliest stages of neuronal development likely represents the beginning of an instructive continuum for the establishment of not only neuronal identity, but also development of sub-circuitry regulating different components of motor behaviors. Here, we show that in the MeA, complementary transcription factor expression marks subsets of progenitors and predicts neuronal subtype identity as defined by molecular and electrophysiological signatures. At the molecular level, *Dbx1*-derived and Foxp2+ neurons express different combinations of the sex steroid hormone pathway protein aromatase and estrogen receptor alpha (ERα). At the electrophysiological level, these two populations possess distinct intrinsic electrophysiological profiles. Thus, our study generates a novel cell-specific transcription factor-based means to predict postnatal MeA neuronal identity.

## Behavioral activation of *Dbx1*-derived and Foxp2+ neurons

The central importance of the MeA for processing innate behaviors such as aggression, mating and predator avoidance is well-established (*Dulac and Wagner, 2006*; *Sokolowski and Corbin, 2012*). Despite this knowledge, the question of which MeA neuronal subtypes encode instinctive behavioral information has only recently begun to be addressed. Recent optogenetic manipulation of the MeA revealed that glutamatergic neurons mediate repetitive self-grooming behaviors while in contrast GABA-ergic neurons regulate either aggressive or mating behaviors, depending on the level of neuronal activity driven by light stimulation (*Hong et al., 2014*). Moreover, MeA neuronal subclasses expressing different components of the stress response system control appropriate behavioral responses to social cues (*Shemesh et al., 2016*). Here, we contribute to the understanding of amygdala cell-specific regulation of behavior by generating a transcription factor based map of MeA

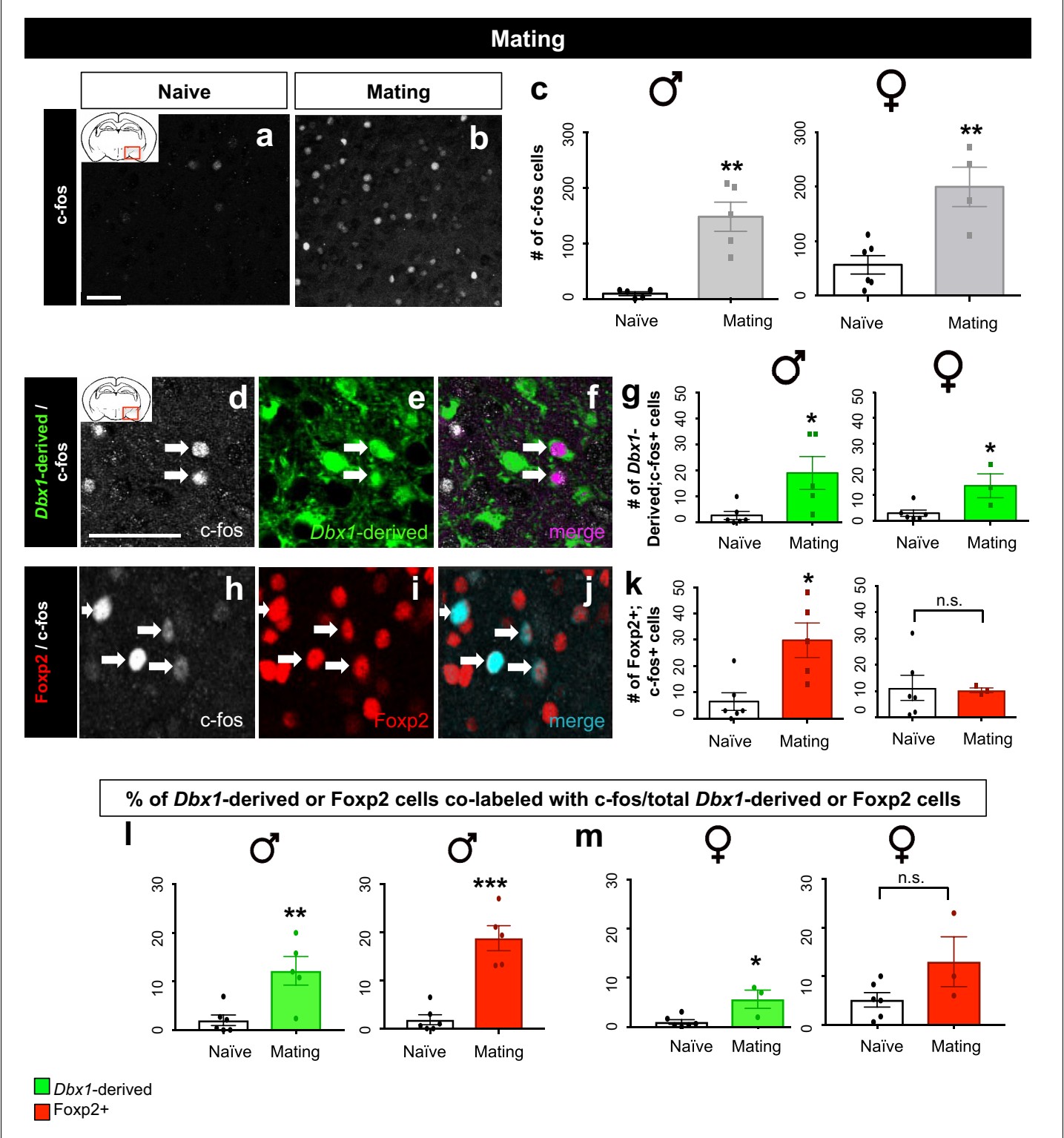

**Figure 6.** Sex-specific subpopulation responses during mating. A significant increase in the number of c-fos+ cells (white) in the MeA during mating compared to naïve control is observed in both male (two tailed Mann-Whitney test, U = 0, p=0.0079, n = 5 naïve, n = 5 mating) and female brains (two-tailed *t*-test, p=0.0038, t = 4.022, df = 8, n = 6 naïve, n = 4 mating) (a–c). Double immunofluorescence for c-fos (white) and YFP (green) or Foxp2 (red) after mating reveals an increase in the number of *Dbx1*-derived neurons expressing c-fos in male (two tailed Mann-Whitney test, U = 2, p=0.0152, n = 6 naïve, n = 5 mating) and female (two tailed Mann-Whitney test, U = 1, p=0.0476, n = 6 naïve, n = 3 mating) brains (d–g). A significant increase in the number of Foxp2+ cells expressing c-fos is only observed in male mice during mating (two-tailed *t*-test, p=0.009, t = 3.331, df = 9, n = 6 naïve, n = 5

*Figure 6 continued on next page*

*Figure 6 continued*

mating) but not in female mice during mating (two tailed *t*-test, p=0.8993, t = 0.1312, df = 7, *n* = 6 naïve, *n* = 3 mating) as compared to naïve controls (h–k). A significant increase in the percentage of *Dbx1*-derived cells expressing c-fos (YFP+ and c-fos+ /total YFP+) is observed in males during mating in comparison to the naïve controls (two-tailed *t*-test, p=0.0067, t = 3.507, df = 9, *n* = 6 naïve, *n* = 5 mating) (l). A significant increase in the percentage of Foxp2+ cells expressing c-fos (Foxp2+ and c-fos+ /total Foxp2+) in male brains during mating is also observed in comparison to naïve controls (two-tailed *t*-test, p=0.0001, t = 6.477, df = 9, *n* = 6 naïve, *n* = 5 mating) (l). An increase in the percentage of *Dbx1*-derived cells expressing c-fos (two-tailed *t*-test, p=0.0131, t = 3.302, df = 7, *n* = 6 naïve, *n* = 3 mating) is observed in female brains during mating (m). In contrast, no increase is observed in the Foxp2+ population (two-tailed *t*-test, p=0.0905, t = 1.96, df = 7, *n* = 6 naïve, *n* = 3 mating) in female brains during mating (m). Data are mean ± s.e.m. *n* is the number of animals. p<0.05 (*), p<0.01 (**), and p<0.001 (***), n.s.; not significant. Arrows show double labeled cells. The scale bar represents 50 μm.

subtype responsiveness to innate-behavioral cues. Thus, our findings provide a developmental molecular context in which to further dissect neuronal subtype control of MeA-driven behaviors.

In addition to playing a central role in processing sensory information required for instinctive behaviors, the MeA is one of the known sexually dimorphic structures of the brain (*Cooke and Woolley, 2005*; *McCarthy and Arnold, 2011*; *Johnson et al., 2008*). Recent studies employing in vivo recording techniques revealed that a significant number of MeA neurons are dedicated to processing olfactory cues from the opposite sex rather than the same-sex, thus providing a direct demonstration of sex-specific differences in sensory processing (*Bergan et al., 2014*). However, the identity of MeA neurons in males and females that differentially process olfactory-based sensory information has not been delineated. Here, we reveal that while *Dbx1*-derived and Foxp2+ neurons are broadly activated by mating, aggressive and predator cues, we found stark differences in *Dbx1*-derived, Foxp2+ and OTP+ cell-specific responses in the male and female brain to mating and predator odor cues. A similar sex-specific control of innate behavior has been directly demonstrated in the ventromedial hypothalamus (VMH), where progesterone receptor-expressing neurons while required for male aggressive and mating behaviors, are only required for female mating behavior (*Yang et al., 2013*). Collectively, our studies in combination with previous studies point to a larger picture in which there are neuronal subpopulations in the MeA and VMH that are involved in the regulation of different innate behaviors in a sex-specific manner.

Although our data do not reveal the neuronal and/or circuit mechanisms underlying our observation of sex-specific subpopulation responses to mating behavior and predator odor presentation, some of our findings do provide potential insight as to how this differential processing may occur. The two most straightforward and non-exclusive potential mechanisms are: 1) with regard to mating, intrinsic differences in *Dbx1*-derived and Foxp2+ neurons and/or 2) subpopulation specific patterns of local and/or long-range connectivity.

Regarding the first potential mechanism, sex steroid hormones and receptors such as aromatase, ERα and AR have been extensively characterized as critical for the output of distinct components of male and female aggressive and mating behaviors (*Yang and Shah, 2014*). For example, deletion of AR resulted in alterations in attack duration during territorial aggression and initiation of male mating (*Juntti et al., 2010*), while ablation of aromatase neurons led to impairments in the production of distinct components of aggression in male and female mice (*Unger et al., 2015*). Consistent with the critical role that these factors play in components of innate behaviors, we found that aromatase is expressed solely in the *Dbx1*-derived lineage. Across species, aromatase has a masculinizing effect (*Wu et al., 2009*; *Balthazart et al., 2011*). Thus, it will be interesting to explore how *Dbx1*-derived aromatase expressing neurons may control male behavioral displays such as mounting and territorial aggression.

Furthermore, both *Dbx1*-derived and Foxp2+ neurons possess distinct electrophysiological profiles, another potential mechanism to control different components of behaviors. Previous work in both vertebrates and invertebrates has revealed that the timing of AP spiking is directly linked to specific behavioral actions. For example, in vertebrates timescale firing differences are associated with dopamine (DA) release for the determination of reward behaviors (*Schultz, 2007*; *Zhang et al., 2009*). It will therefore prove interesting to explore if and how firing patterns of *Dbx1*-derived and Foxp2+ MeA neurons may control different components of innate behaviors across sexes.

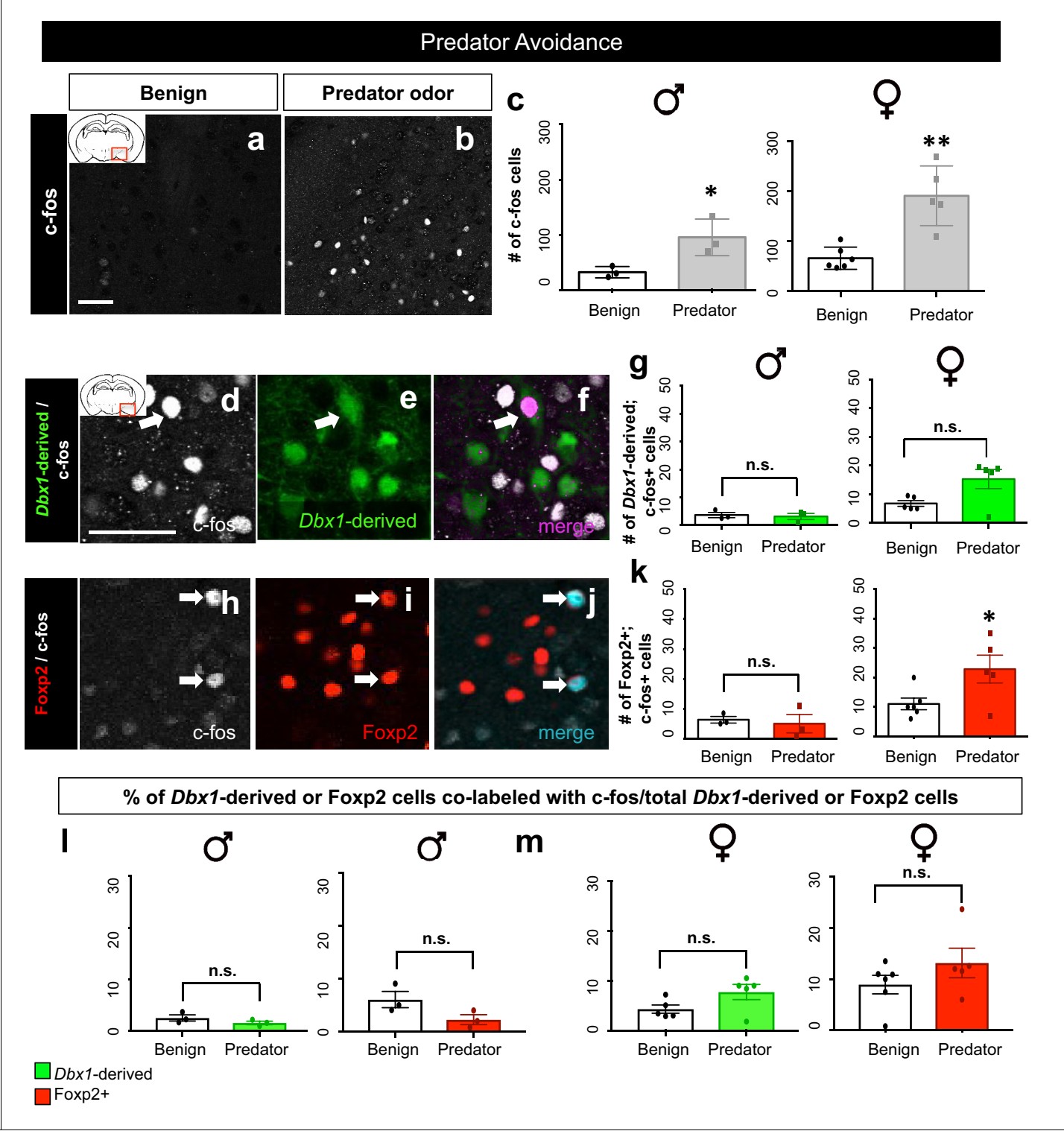

**Figure 7.** *Dbx1*-derived and Foxp2+ MeA subpopulation activation patterns in response to predator odor. A significant increase in the number of c-fos + cells (white) in the MeA in both male (two-tailed *t*-test, p=0.0344, t = 0.30.153, df = 4, *n* = 3 benign, *n* = 3 predator avoidance) and female (two-tailed *t*-test, p=0.001, t = 4.778, df = 9, *n* = 6 benign, *n* = 5 predator avoidance) brains is observed in the presence of rat bedding compared to benign unsoiled bedding (a–c). Double immunofluorescence for c-fos (white) and YFP (green) or Foxp2 (red) after predator odor exposure reveals no increase in the number of *Dbx1*-derived cells (two-tailed *t*-test, p=0.687, t = 0.35, df = 4, *n* = 3 benign, *n* = 3 predator avoidance) (d–g) or Foxp2+ cells (two-tailed *t*-test, p=0.703, t = 0.70, df = 4, *n* = 3 benign, *n* = 3 predator avoidance) (h–k) expressing c-fos in male brains as compared to control. In the female brain, there is also no increase in the number of *Dbx1*-derived cells during predator odor exposure as compared to control (two tailed Mann-

*Figure 7 continued on next page*

*Figure 7 continued*

Whitney, U = 5, p=0.1349, *n* = 5 benign, *n* = 5 predator avoidance) (**d–g**). In contrast, an increase the number of Foxp2+ cells (two-tailed t-test, p=0.036, t = 2.47. df = 9, *n* = 6 benign, *n* = 5 predator avoidance) in female brains is observed (**h–k**). Analysis of the percentage of *Dbx1*-derived (YFP+ and c-fos+ /total YFP+) and Foxp2+ cells (Foxp2+ and c-fos+/ total Foxp2+) expressing c-fos in both male and female brains revealed no increases in either population (*Dbx1*-derived population in male mice: two-tailed *t*-test, p=0.2318, t = 1.408, df = 4, naïve *n* = 3, predator avoidance *n* = 3; Foxp2+ population in male mice: two-tailed *t*-test, p=0.1023, t = 2.11, df = 4, naïve *n* = 3, predator avoidance *n* = 3; *Dbx1*-derived population in female mice: two-tailed Mann-Whitney, U = 5, p=0.1349, naïve *n* = 5, predator avoidance *n* = 5; Foxp2+ population in female mice: two-tailed *t*-test, p=0.2299, t = 1.288, df = 9, naïve *n* = 6, predator avoidance *n* = 5). Data are mean ± s.e.m. *n* is the number of animals. Arrows show double-labeled cells. p<0.05 (*), p<0.01 (**), and p<0.001 (***), n.s.; not significant). The scale bar represents 50 μm.

The following figure supplement is available for figure 7:

**Figure supplement 1.** Sex-specific activation of OTP+ cells during predator avoidance.

The second potential and perhaps more intriguing mechanism that may account for *Dbx1*-derived and Foxp2+ subtype specific male versus female patterns of neuronal activation during mating are sex specific local and/or long-range patterns of connectivity. Although currently not yet observed in a brain circuit in mammals, such a mechanism has recently been uncovered in *c. elegans* in which shared male and female circuits show differences in connectivity that is established during wiring (*Oren-Suissa et al., 2016*). Consistent with this, there is some suggestion of sex-specific differences in olfactory-MeA projections in rodents (*Kang et al., 2011*). Although we did not differentiate according to the sex of the animal, we found differences in both the amplitude and frequency of EPSCs between lineages, indicating differences in the strength and/or number of inputs between *Dbx1*-derived and Foxp2+ MeA neurons. Although the source of input cannot be determined from

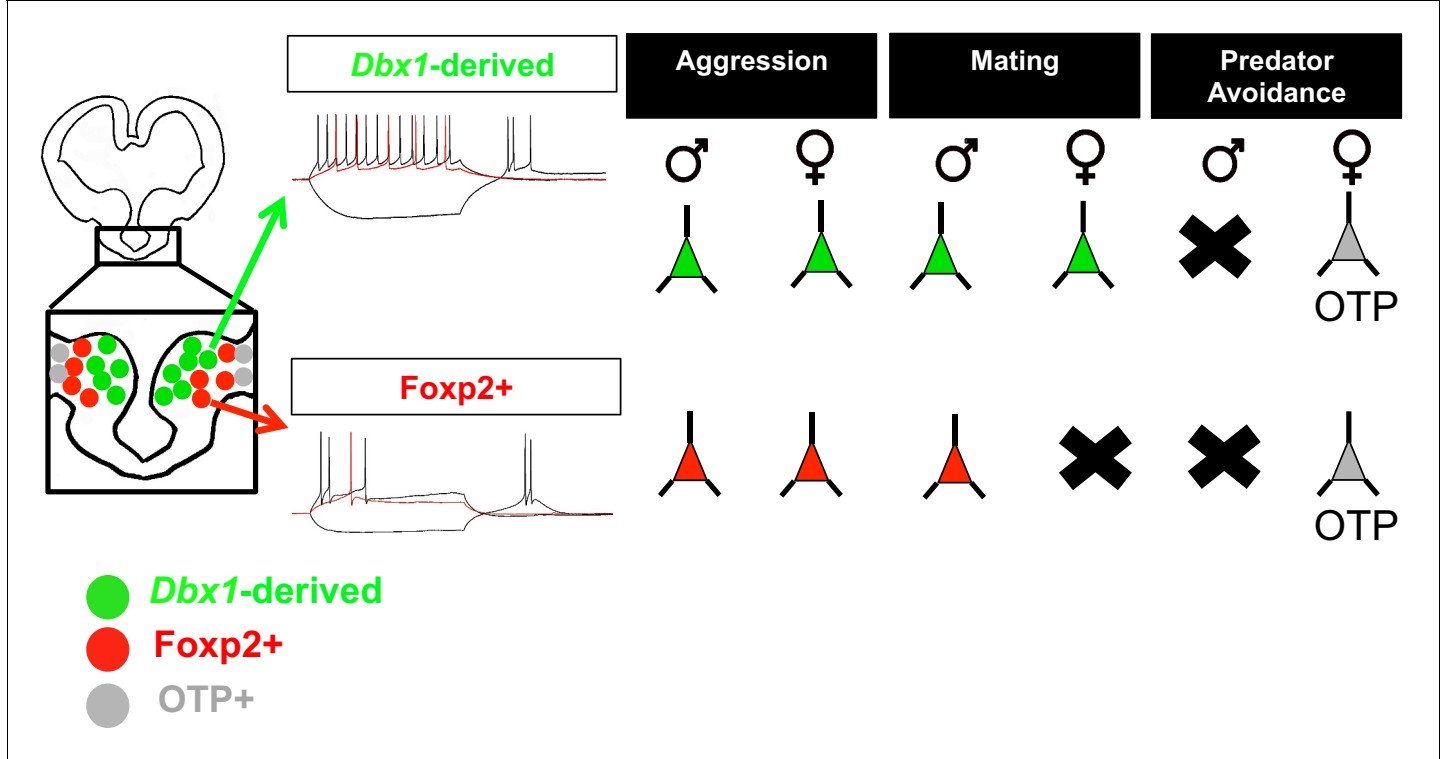

**Figure 8.** Summary of findings. Schematic summarizing our findings in which embryonic segregation of *Dbx1*-derived (green) and Foxp2+ (red) cells in the embryonic brain predicts postnatal intrinsic electrophysiological profiles and differential sex and subpopulation-specific neuronal activation patterns (neurons indicate activation and X's indicate no activation) in response to aggression (territorial and maternal), mating and predator avoidance. The OTP+ population is shown responding to predator odor in female brains. The male predator odor responsive population remains to be determined.

our analysis, there are direct excitatory inputs to the MeA emanating from the mitral/tufted neurons of the accessory olfactory bulb (AOB) (*Martel and Baum, 2009*; *Bergan et al., 2014*). Thus, determination of input/output wiring patterns of male and female *Dbx1*-derived and Foxp2+ MeA neurons will be highly informative.

In summary, although the precise instructive developmental mechanisms programming innate behaviors remain to be elucidated, we reveal that differential transcription factor expression during development is predictive of neuronal identity based on molecular and electrophysiological criteria and sex-specific patterns of neuronal activation during innate behaviors.

# Materials and methods

## Animals

Mice were housed in the temperature and light-controlled (12 hr light-dark cycle) Children's National Medical Center animal care facility and given food and water *ad libitum*. All animal procedures were approved by the Children's National Medical Center's Institutional Animal Care and Use Committee (IACUC) and conformed to NIH Guidelines for animal use. Mice used were *Dbx1cre*$^{+/-}$ (kindly provided by A. Peirani, Institut Jacques Monod, Paris), *Shhcre*$^{+/-}$ (Jackson Labs strain B6.Cg-*Shhtm1 (EGFP/cre)Cjt*/J), *Emx1cre*$^{+/-}$ (Jackson Labs strain B6.129S2-*Emx1tm1(cre)Krj*/J), *Nkx6.2cre*$^{+/-}$(Jackson Lab strain Tg *Nkx6-2-icre)1Kess/SshiJ*), *Foxp2cre*$^{+/-}$(kindly provided by R. Palmiter, University of Washington) (*Rousso et al., 2016*), all crossed to *Rosa26YFP*$^{+/+}$mice (Jackson lab strain R26R-EYFP). For analysis of aromatase expression, we used Aromatase LacZ reporter mice (kindly provided by N. Shah, University of California-San Francisco) (*Wu et al., 2009*). Mice were genotyped by Transnetyx Inc. Genotyping Services. Adult mice were considered between 3–7 months of age. The sample size was based on previous experiments and published data. No statistical methods were used to determine sample sizes.

## Immunohistochemistry

Postnatal mice were transcardially perfused with 4% paraformaldehyde (PFA) and brains post fixed overnight at 4°C, embedded in 4% agarose (Invitrogen) and sectioned at 50 µm with a vibrating microtome (Leica VT1000S). Embryos were fixed in 4% PFA overnight at 4°C degrees, cryoprotected in 30% sucrose, embedded in O.C.T. compound (Tekka) and sectioned at 20 µm on a cryostat (Leica CM1850). For IHC, tissue sections were incubated overnight with primary antibody, then washed with PBST and 10% normal donkey serum and incubated for 4 hr with the corresponding secondary antibodies, and mounted with DAPI Fluoromount (SouthernBiotech 0100–20, Birmingham, AL). Primary antibodies used were rat anti-GFP (to detect YFP expression, (1:1000, Nacalai 04404–84, San Diego, CA), goat anti-Foxp2 (1:200; Santa Cruz sc-21069, Dallas, TX), rabbit anti-Foxp2 (1:500; abcam ab16046, Cambridge, UK), rabbit anti-OTP (1:2000; kind provided by F. Vaccarino, Yale University), rabbit anti-androgen receptor (1:750; Epitomics AC-0071, Cambridge, UK), rabbit anti-estrogen receptor α (1:6000; Millipore 06–935); mouse anti-NeuN (1:200; Millipore MAB-377), mouse anti-CC1 (1:250; Calbiochem OP80-100), goat anti-calbindin (1:200; Santa Cruz sc-7691), rat anti-somatostatin (1:100; Millipore MAB354, Billerica, MA), rabbit anti-cfos (1:500; Santa Cruz sc-52, Dallas, TX), goat anti-cfos (1:300; Santa Cruz sc-52G, Dallas, TX), anti-rabbit nNOS (1:8000; ImmunoStar 24287, Hudson, WI), mouse anti-CAMKIIα (1:500 Biomol ARG22260.50, Farmingdale, NY) and chicken anti-$\beta$Gal (1:2000; abcam 9361, Cambridge, UK). Secondary antisera used were donkey anti-rat or anti-goat Alexa 488 (1:200; Life Technologies, Waltham, MA), anti-rabbit or anti-goat Cy5 (1:1000; Jackson ImmunoResearch, Westgrove, PA), anti-rabbit or anti-mouse Cy3 (1:1000; Jackson ImmunoResearch, Westgrove, PA), anti-mouse dylight 649 (1:500; Jackson ImmunoResearch, Westgrove, PA), or anti-chicken dylight 549 (1:500; Jackson ImmunoResearch, Westgrove, PA).

## Microscopy

Fluorescent photographs were taken using an Olympus FX1000 Fluoview Laser Scanning Confocal Microscope (1 um optical thickness).

## Quantification

### Molecular marker analysis

For embryonic analysis, the average of 2–3 sections encompassing the POA were imaged and quantified (*Figure 1*; *Figure 1—figure supplements 1–2*). For adult analyses (*Figures 1–7*; *Figure 1—figure supplements 1–2*; *Figure 2—figure supplement 1*; *Figure 4—figure supplement 1*; *Figure 5—figure supplement 1*; *Figure 7—figure supplement 1*), every sixth serial coronal section encompassing the anterior to posterior extent of the MeA (Bregma −1.30 to −1.90, see *Figure 1—figure supplement 3*) was immunostained with antibodies to Foxp2, YFP and markers of interest. Quantification was done by counting single and double-labeled cells encompassing the entire domain of expression within the POA or MeA (*Figures 1–2* and *4*; *Figure 1—figure supplements 1–2*; *Figure 2—figure supplement 1*; *Figure 4—figure supplement 1*).

### Neuronal activation

For analyses of neuronal activation (*Figures 5–7*, *Figure 7—figure supplement 1*), a single MeA section with the highest number of c-fos+ cells corresponding to the presence of *Dbx1*-derived, Foxp2 + or OTP+ cells was chosen for quantification. c-fos, YFP and Foxp2 triple immunostaining was conducted on the same section and single and double-labeled cells counted. c-fos and OTP double immunostaining was conducted and single and double cells counted.

## Statistical evaluation

Unless otherwise stated, data were analyzed using GraphPad six statistical software. We first tested the distribution of the data with the Shapiro-Wallis test for normality. Data that were normally distributed was analyzed using an unpaired two-tailed *t*-test for analysis of experiments involving two groups (*Figure 1—figure supplement 1*; *Figure 1—figure supplement 2*; *Figure 5x*; *Figure 6c* females, k, m, l; *Figure 7c,g* males, k, l, m Foxp2+ subpopulation; *Figure 7—figure supplement 1c, d,h,i*) and a one-way ANOVA (*Figure 5o,w,y* Foxp2+ subpopulation) followed by Tukey-Kramer multiple comparison test was used for analysis of experiments involving three or more groups. Data with a non-normal distribution were analyzed by using the non-parametric test Mann-Whitney when comparing two groups (*Figure 5c,g,k,y* *Dbx1*-derived subpopulation; *Figure 6c* males, g; *Figure 7g* females, m *Dbx1*-derived subpopulation; *Figure 7—figure supplement 1j,k*) and Kruskal-Wallis with Dunn's post-hoc corrections for data with three groups (*Figure 5—figure supplement 5s*). For the analysis of data shown in *Figure 4* and *Figure 4—figure supplement 1* we performed the following statistical analysis: when the data met the normality assumption or could be transformed to meet the normality assumption, generally two-way analysis of variance models were implemented to evaluate the evidence of differences in mean effects of the two experimental factors on cell activation (*Figure 4g,u*; *Figure 4—figure supplement 2a,b*). In the situation where no data transformation could be found to achieve an acceptable level of normality, quantile regression was performed, which does not require the normality assumption, to evaluate comparable differences in median effects (*Figure 4n*; *Figure 4—figure supplement 2c*). In each case, the initial models included a cross-products term to assess evidence of effect modification or interaction. When there was no interaction, it was taken as evidence of the absence of effect modification and the cross-products term was removed leaving only a model that assessed independent effects of the each factor separately, while holding constant any effects of the other factor. Depending on the underlying model, either mean or median effects ± 95% confidence intervals were derived to reflect the differences that were consistent with statistically meaningful differences in the final model. Under consideration of protecting the experiment-wise error rate, as long as the evaluation of differences focused only on identifying the nature of effects deemed statistically meaningful in the final model, there was no correction made for multiple comparisons. Analysis of data meeting the normality assumption was based on GraphPad Prism six and analysis based on quantile regression was implemented in Stata 13. As mice had to be sacrificed after each behavioral assay in order to conduct c-fos immuno-analysis, technical repeats were not available. Measurements from different mice were considered biological repeats to determine sample size. Data points were considered outliers and excluded if they were two standard deviations away from the mean.

## Electrophysiology and biocytin filling

Mice (P25-40) were anaesthetized with isoflurane and sacrificed. Brains were removed and immediately immersed in ice-cold oxygenated (95% O2/5% CO$_2$) sucrose solution (234 mM sucrose, 11 mM glucose, 26 mM NaHCO$_3$, 2.5 mM KCl,1.25 mM NaH$_2$PO$_4$.H$_2$O, 10 mM MgSO$_4$ and 0.5 mM CaCl$_2$). Coronal slices of 300 μm in thickness were cut. Slices with amygdala were collected and placed in oxygen-equilibrated artificial cerebral spinal fluid (ACSF) as previously described (*Hirata et al., 2009*). Either *Dbx1*-derived or *Foxp2*-derived neurons were then visualized using a fluorescent lamp (Nikon) with a 450-490λ filter. Whole-cell patch-clamp recordings from YFP-positive fluorescent cells were performed at room temperature with continuous perfusion of ACSF (Multiclamp 700A, Digi-DATA1322, Molecular Devices). Intracellular solution (in mM): 130 Kgluconate, 10 KCl, 2 MgCl2, 10 HEPES, 10 EGTA, 2 Na2-ATP, 0.5 Na2-GTP. All measurements of intrinsic and synaptic properties were analyzed off-line using Clampfit Software (V.10.2, Molecular Devices) and graphing software (OriginPro 9.1). At the end of each recording, biocytin (1%) was injected with the depolarizing current (1nA) for post-hoc morphology analysis. All slices were then fixed with paraformaldehyde overnight at 4°C and processed for Fluorescein-conjugated Avidin-D (1:200, Vector Laboratories), YFP IHC (*Dbx1*-derived and Foxp2+ recordings) or Foxp2 IHC (for Foxp2+ recordings) as described above.

## Neuronal reconstruction

Neurons were filled with biocytin and imaged using an Olympus FX1000 Fluoview Laser Scanning Confocal Microscope (0.5 um optical thickness). VIAS software was used to align confocal images taken at 40x and 60x in the same plane (x,y,z). Neurons were then traced using neuTube software, which uses fixed radii small tubes to estimate the dendritic branches length and thickness (*Feng et al., 2015*).

## Behavioral assays

*Dbx1cre$^{+/-}$;Rosa26YFP$^{+/+}$* male and female mice 3–7 months old were used for the behavioral assays. One week prior to testing, animals were single housed. Testing was performed between the hours of 18:00 and 20:00 corresponding to the beginning of the dark cycle for all assays except the maternal aggression assay which took place from 13:00 to 15:00pm corresponding to the light cycle.

### Mating

Sexually naïve hormonally primed females were analyzed during female mating. Mating was assessed by placing a mouse of the opposite sex inside the resident's cage and checking for plugs every 30 min. When a plug was observed, noting successful intromission, female or male mate was removed from the cage and the experimental animal was left inside the cage for an additional 30 min before being sacrificed. Females with no plugs were excluded from the analysis as no mating occurred.

### Territorial and maternal aggression

Male territorial aggression was assessed by performing a resident-intruder assay in which an unfamiliar male mouse ('intruder') was placed inside the resident's cage for 10 min. During this period, the homecage male displayed typical aggressive behaviors including attacking and biting. The intruder was removed and then after an additional 50 min the resident male was sacrificed. For the maternal aggression assay, female mice were single-housed after a plug was observed. The following experimental conditions were run: (1) for maternal aggression pups age between P5-P8 were removed and 2 min later a sexually naïve male was introduced into the cage for 10 min, (2) for 'no pups' condition, pups were removed but no intruder presented and (3) for naïve control, the cage was undisturbed in which pups were not removed and no intruder presented. In conditions (1) and (2) the female mice were sacrificed after 50 min after pup removal. If female mice did not actively attack the intruder at least two times, they were excluded from the analysis. All male mice attacked the intruder for the 10 min period.

### Predator avoidance

Predator avoidance was assessed by introducing a petri dish containing rat bedding to the homecage for 1 hr. The control predator group was presented with a petri dish containing clean mouse

bedding (benign). Mice were sacrificed after 1 hr presentation of rat bedding or benign bedding. Mice that did not show escaping responses after the presentation of the petri dish were excluded.

## Acknowledgements

We thank V Gallo, J Triplett and I Zohn for constructive input and/or critical reading of the manuscript. We also thank members of the Corbin and Triplett labs for input during the course of this study and Robert McCarter, Director of the DC- IDDRC Biostatistics and Informatics Core for his expert biostatistics assistance. We thank A Pierini for *Dbx1^{cre}* mice, R Palmiter for *Foxp2^{cre}* mice sent prior to publication (*Rousso et al., 2016*), N Shah for *AromataseLacZ* reporter mice, and F Vaccarino for the OTP antibody. This work was partially supported by NIH grants R01 NIDA020140 (JGC), and R01 DC012050 (JGC). Core support was provided by the CNMC DC-IDDRC Imaging, Biostatistics and Informatics and Animal Neurobehavior Evaluation Cores (NIH IDDRC P30HD040677). JEL is a predoctoral student in the Molecular Medicine Program of the Institute for Biomedical Sciences at The George Washington University. This work is from a dissertation to be presented to the above program in partial fulfillment of the requirements for the Ph.D. degree.

## Additional information

### Funding

| Funder | Grant reference number | Author |
| --- | --- | --- |
| National Institute on Drug Abuse | F32 DA035754 | Katie Sokolowski |
| Goldwin Foundation Grant for Pediatric Epilepsy | | Judy Liu |
| National Institute on Drug Abuse | R01 NIDA020140 | Joshua G Corbin |
| National Institute on Deafness and Other Communication Disorders | R01 DC012050 | Joshua G Corbin |

The funders had no role in study design, data collection and interpretation, or the decision to submit the work for publication.

### Author contributions

JEL, Conceptualization, Data curation, Formal analysis, Supervision, Validation, Investigation, Visualization, Methodology, Writing—original draft, Writing—review and editing; KS, SE, Data curation, Formal analysis, Supervision, Validation, Investigation, Methodology; PL, Data curation, Formal analysis, Validation, Investigation, Visualization, Methodology; YK, Data curation, Formal analysis, Investigation, Methodology; MG, Data curation, Formal analysis, Supervision, Validation, Investigation, Methodology, Writing—review and editing; LO, Data curation, Formal analysis, Validation, Investigation, Methodology, Writing—review and editing; TRH, MK, DF, Data curation, Investigation; MH, Conceptualization, Supervision, Methodology; JL, Supervision, Funding acquisition; JGC, Conceptualization, Data curation, Formal analysis, Supervision, Funding acquisition, Validation, Investigation, Visualization, Methodology, Writing—original draft, Project administration, Writing—review and editing

### Author ORCIDs

Julieta E Lischinsky, http://orcid.org/0000-0003-1664-6642
Joshua G Corbin, http://orcid.org/0000-0003-0122-4324

### Ethics

Animal experimentation: All animal procedures were approved by the Children's National Medical Center's Institutional Animal Care (Animal Welfare Assurance Number: A3338-01) and Use Committee (IACUC) protocols (#00030435) and conformed to NIH Guidelines for animal use. All surgery was

performed under ketamine/xylazine cocktail anesthesia, and every effort was made to minimize suffering.

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
