## [Decision Letter]

Thank you for submitting your article "Embryonic lineage predicts medial amygdala neuronal identity and sex-specific responses to innate behavioral cues" for consideration by *eLife*. Your article has been reviewed by three peer reviewers, and the evaluation has been overseen by a Reviewing Editor and a Senior Editor. The following individuals involved in review of your submission have agreed to reveal their identity: Margaret M McCarthy (Reviewer #1); Dayu Lin (Reviewer #3).

The reviewers have discussed the reviews with one another and the Reviewing Editor has drafted this decision to help you prepare a revised submission.

Summary:

The reviewers were enthusiastic about your study linking embryonic patterning of the medial amygdala, neuronal identity and amygdala-directed instinctive behaviors, calling it "exciting" and "elegant". The study nicely charts the lineage of two distinct cell types that originate in the POA and then populate the medial amygdala, and shows that these cell populations do not overlap, have distinct electrophysiological profiles, and respond to varying degrees in males versus females in tests of aggressive, sexual and fear behavior.

Essential revisions:

A major concern of reviewers 2 and 3 is that the quantification of the different populations (Dbx1, OTP, Foxp2) and the modes of representation are confusing. For example, 15% overlap of all 3 markers is unclear, especially if, as reviewer 2 says, 85% OTP cells are neither Foxp2+ nor Dbx1 derived, this reviewer asks why this population seems to be ignored in subsequent electrophysiological and behavioral analyses? Please clarify or defend this decision.

Similarly, reviewer 2 appreciated that during mating behavior, the Dbx1 population showed increased c-fos expression in males and females but Foxp2 cells increased their c-fos in males only. Are you able to address which populations account for the increase in c-fos in predator behavior in males, and whether these cells could be the OTP cells?

Reviewer 3 believes that it would be helpful to indicate the percentage of OPT+ and Dbx1+ cells, and the expected overlap by random chance, even though Foxp2+ and OPT+ do seem to be largely non-overlapping as they occupy virtually non-overlapping spatial locations.

Other comments that could be addressed textually:

Reviewer 1 requests being mindful of the concept of sex differences (in c-fos expression), but not terming these sexual dimorphisms, even though reviewer 2 speaks of "demonstrate sexually dimorphic activation of specific lineage-derived populations in the medial amygdala".

This reviewer also called for citing a previous study on the dissonance between estrogen receptor and ligand, along with different expression of aromatase.

The reviewers thought that Figure 8 was too speculative and that Figure 8—figure supplement 1 be used instead.

Reviewer #1:

This is an exciting paper. The authors trace the lineage of two distinct cell types that originate in the POA and then populate the medial amygdala. Both cells are inhibitory but one is derived from the cells that originally expressed the brain development gene Dbx1, while the other expresses the transcription factor Foxp2 throughout life. These cells do not overlap, show distinct electrophysiological profiles and respond to varying degrees in males versus females in tests of aggressive, sexual and fear behavior. The anatomy is exquisite and the execution of the behavioral tests and conclusions reached are well grounded in the data. Only modest changes are needed, mostly in terminology, and a few points for pondering are offered.

1) The authors make the common and frequent mistake of referring to even a modest sex difference as a sexual dimorphism. It may seem like semantics but the term sexual dimorphism should be reserved for those sex differences which involve two (di-) forms (morphs). At this point, what the authors have discovered is a sex difference in c-fos expression, not a sex dimorphism. This is no less interesting. I suggest the authors comb through the manuscript and at each point where they have used the term sexual dimorphism ask if sex difference would serve the same purpose.

2) The authors also report a fascinating difference in the cellular distribution of aromatase, with about half of Dbx- lineage cells expressing it but Foxp2 cells not at all. There were no cell lineage effects observed for AR or ER. This is consistent with the notion that sexual differentiation of the brain is mediated by a sex difference in ligand, i.e. estradiol, and not by the amount or cellular localization of the receptor. The functional significance of having aromatase in some cells that express ER and others that do not is unknown but the authors might consult work on the Japanese quail brain and the analogous POA (POM) where a similar circumstance is found.

Reviewer #2:

Lischinsky et al. present an elegant story in which they demonstrate sexually dimorphic activation of specific lineage-derived populations in the medial amygdala. Their data are clear and approach logical, and the results are exciting. A couple of major concerns are detailed below.

1) The text is difficult to follow in some important places. For example, looking at Figure 1—figure supplement 1, and the imprecise text associated with it ("generally distinct,", subsection “Dbx1 and Foxp2 expression segregates embryonic and postnatal MeA subpopulations”) one does not immediately grasp whether there are three populations (Foxp2, OTP, and Dbx derived) or two. The graph in f is very confusing- the X axis appears to be the fraction of OTP cells that are Dbx1 derived, plotted against% colocalization with Foxp2, which indicates a 15% overlap of all 3 markers but surely there's an easier way to present this.

This becomes important because if 85% OTP cells are neither Foxp2+ nor Dbx1 derived, then why do the authors ignore this population in their subsequent electrophysiological and behavioural analysis? The study seems incomplete given the presence of this population. If the authors provide an explanation for why they do not examine this population further, that would be sufficient, but it needs to be well justified.

2) The examination of which population is active in different behaviors is beautifully done and the most interesting part of the paper. During mating behavior, c-fos increased in the Dbx1 derived population in both males and females, but in the Foxp2 population, c-fos increase was seen only in males. This is a key result. In predator avoidance, however, the results are puzzling. On the one hand, both males and females show an increased c-fos expression in the MeA. Yet when the Dbx1 derived and Foxp2+ population was examined, it showed an increase only in females and not in males. The authors do not examine which cells account for the c-fos increase in males. Could this perhaps be the OTP population (point 1).

3) The schematic in Figure 8 is highly speculative and goes beyond the data. The schematic seeks to imply that Foxp2+ neurons participate in mounting behaviour, which females do not display, which perhaps explains why this population appears silent during mating in females. This is needlessly speculative- such a hypothesis would need to be examined very carefully and I suggest the schematic be eliminated. In Figure 8, the authors suggest a sexually dimorphic connectivity such that Foxp2+ neurons are inhibited only in females but not in males. There is no evidence in support of this either. I suggest instead that the summary schematic in Figure 8—figure supplement 1 should be used as Figure 8, it helps the reader.

Reviewer #3:

In Lischinsky et.al. study, the authors identified two nearly exclusive populations in the MeA. One is Dbx1 derived as previously described and the other is Foxp2 derived. These two populations are distinctive at both embryonic and adult stage and have differential electrophysiological properties. The authors also investigated the Fos induction pattern after various social behaviors and concluded that the Foxp2 population is involved in only male mating not female mating, while aggression, predator odor induced similar responses in Dbx-1 and Foxp2 derived cells in both males and females.

The study is overall well performed. The finding is novel and interesting. The developmental origin of diverse MeA cells is poorly understood. This paper represents an important addition to our knowledge. However, the cell counting analysis is incomplete and the images could be improved. As a result, some important information might be missed in the current paper.

1) When understanding the overlap between two types of cells, the absolute number is not very informative. The percentage of cells of double labeled cells over single labeled cells provides some understanding of the overlap but it does not address the real question – are the two populations under investigation preferentially overlapped or not? For example, imagine that Dxb1 population accounts for 50% of the total number of cells and the Esr1 accounts for 50% of the total number of cells, if the double labeled cell is 25% of Dbx1 cells, then the Esr1+ cells are preferentially not derived from the Dbx1 cells. However, if Esr1 account for 10% of the total population, then the Esr1 is preferentially derived from the Dbx1 cells. Without knowing the percentage of the Esr1 cells, it is unclear what the 25% overlap actually means. Thus, throughout the paper, the authors should include information regarding the percentage of cells labeled by gene1, the percentage of cells labeled by gene 2 and the percentage of double labeled cells. Whether the two genes are preferentially labeled or not could be calculated as double%/(Gene1% xGene2%). A value over 1 means that the two genes are preferentially overlapped and a value below one will mean the opposite. With multiple animals, it can be calculated whether the preferential (or non-preferential) overlap is statistically significant or not.

2) While the authors analyzed the sexually dimorphic involvement of the Dbx1 and Foxp2 cells in social behaviors, the basic characterization of the Dbx1 and Foxp2 cells in males and females is lacking. Does the percentage or the absolute number of Dbx1 or Foxp2 cells differ between sexes?

3) Regarding the overlapping of the Dbx1/Foxp2 with behavior induced Fos, there are several questions that could be readily addressed with the data. First, is the Dbx1 or Foxp2 cells preferentially/non-preferentially activated by the behavior (e.g. aggression)? For example, in Figure 5, the number of Dbx-1 cells expressing aggression induced Fos appears to be quite low. It is about 5% of the total Fos cells based on the graph. Although the Dbx1% is not reported, based on Figure 1, it appears to be quite dense, probably over 20% of the total cells. If so, the Dbx1 cells seem to be preferentially not activated during aggression. This should be analyzed and addressed. Second, to address whether the overlap between the Dbx1/Foxp2 and Fos is sexually dimorphic or not, it is better to compare the value of double%/(Gene1% xGene2%) rather than the double%. The double% is less informative as this value will be affected by the Fos% and the Dbx1% or FoxP2%. If Fos% is sexually dimorphic to begin with, it does not mean much if the double% differs between sexes.

4) Given the different number of Fos cells in Foxp2 and Dbx1 populations (e.g. Figure 5), it is likely that the distribution of Fos is biased topographically in the MeA. It will be helpful to present large view images of the Fos induction patterns. This will help the readers to understand potential reasons for overlap/non-overlap.

5) The statistics in the paper should be double checked or the number of animals needs to be increased to provide sufficient statistical power. For example, in Figure 4, there is a strong trend (if not yet significant) that the co-labeled% in female Dbx-1 population is higher than the rest of the groups. If one perform a t-test between the male and female Esr1 and Dbx1 double labeled%, it should be significant given that there is no overlap in values and the variability is not huge.

6) Lastly, the authors should provide the Figure number in each figure file to facilitate the review process.

Overall, this is a good study with interesting findings. However, the results can be better analyzed to better understand the potential molecular features and behavioral relevance of Dbx1 and FoxP MeA cells.

[Editors' note: further revisions were requested prior to acceptance, as described below.]

Thank you for resubmitting your work entitled "Embryonic transcription factor expression in mice predicts medial amygdala neuronal identity and sex-specific responses to innate behavioral cues" for further consideration at *eLife*. Your revised article has been favorably evaluated by a Senior Editor, a Reviewing Editor, and one reviewer. The second reviewer had added comments that were in agreement with the reviewer 1.

You have done an excellent job of responding to the reviews of your revised manuscript with added additional data and new analyses. The study indeed contributes to an understanding of the developmental logic for processing of innate behaviors by different MeA neuronal subtypes. However, there are some remaining issues that need to be addressed before acceptance, as outlined below:

1) You changed the statistical tests in new Figure 4:

You state in response to Reviewer #3 Critique #5, that some groups in Figure 4 looked significantly different, but no such difference was detected by ANOVA, and then you proceeded to conduct 2-tailed t-tests. The reviewers believe that there is no justification for this approach, as one of the recurrent themes of the manuscript is the presence and absence of sex differences. Neither can be concluded when males and females are analyzed separately, only within sex comparisons can be made. In Figure 4 the bars indicate a significant difference between males and females but this panel alone involves at least 3 t-tests, which is an inappropriate use of multiple comparisons without correction. This is true for most of the data in this figure, and perhaps others. The appropriate analysis is an ANOVA with sex as a factor and only when there is either a main effect of sex or a sex by other factor interaction can post-hoc tests be done.

Further, for use of one way ANOVA, the data need to be normally distributed and exclude outliers. One outlier can throw off the statistics and provide a false negative. While pairwise comparison should not be done without finding significance in ANOVA, it is essential to check whether ANOVA is correctly used here.

Given the very small n's in some of these endpoints, which is understandable given the labor intensiveness of the co-localization analyses, it is imperative that correct statistics are conducted. In this instance it is probably better to miss a sex difference or effect than it is to incorrectly claim one based on a small n and inappropriate statistics. We thus encourage you to seek advice from someone well-versed in statistical analysis to help you work through this, report back to us, and revise the manuscript accordingly.

2) Also in relation to the graphical presentation of statistics: in Figure 5, and others, the use of bars above the histograms is confusing as usually an open-ended bar indicates all groups below it are different from each other. There are clearer ways to indicate post-hoc significant differences between 2 groups, but hard to explain in writing: either a combination of letters, or bars with end feet that point to only the 2 groups. You are encouraged to look at other papers with ANOVA's for some ideas.

3) The reviewers appreciate your removal of the mis-used term "lineage" from your title, changed to: "Embryonic transcription factor expression in mice predicts medial amygdala neuronal identity and sex specific responses to innate behavioral cues".

---

## [Author Response]

Essential revisions:

A major concern of reviewers 2 and 3 is that the quantification of the different populations (Dbx1, OTP, Foxp2) and the modes of representation are confusing. For example, 15% overlap of all 3 markers is unclear, especially if, as reviewer 2 says, 85% OTP cells are neither Foxp2+ nor Dbx1 derived, this reviewer asks why this population seems to be ignored in subsequent electrophysiological and behavioral analyses? Please clarify or defend this decision.

In previous Figure 1—figure supplement 1, we mislabeled the y-axis, which lead to some confusion. The axes are now correctly labeled (revised Figure 1—figure supplement 1).

Our focus was on Dbx1-derived and Foxp2+ subpopulations primarily due to our observations that both populations mark the POA, the telencephalic niche that we were most interested in studying.

Additionally, the availability of both *Dbx1^cre^* and *Foxp2^cre^* mice greatly facilitated our technical ability to conduct patch clamp recordings specifically in these two populations. To our knowledge, OTPcre mice have yet to be generated. However, the reviewers raise an interesting and important point that the OTP population may be of interest. To further investigate the identity of these neurons, following the reviewers’ input we conducted a number of new experiments:

1) To examine in greater detail the potential overlap with the OTP+ population and the Dbx1-derived and Foxp2+ populations we conducted a new series of double and triple immuno-labeling at postnatal stages. We found that Dbx1-derived and Foxp2+ populations were generally not overlapping with the OTP+ cells. This new data is shown in new Figure 1—figure supplement 1.

2) We further conducted a series of double immuno-labeling experiments with a series of cell subtype markers to examine the identity of OTP+ cells. Previous published analyses (Garcia-Moreno et al. Nature Neuroscience, 2010) suggested that a subpopulation of OTP neurons may be excitatory. However, this analysis was cursory and done at embryonic stages. Therefore, to assess the identity of this population in more detail, we performed double immuno-staining with OTP and: YFP+ (in *Emx1^cre^;RYFP* mice which mark excitatory neurons), CAMKIIα (excitatory neurons), calbindin (inhibitory neurons) and somatostatin (inhibitory neurons). From these new analyses, we found that the majority of OTP+ neurons are inhibitory. This new data is shown in new Figure 2—figure supplement 1.

Similarly, reviewer 2 appreciated that during mating behavior, the Dbx1 population showed increased c-fos expression in males and females but Foxp2 cells increased their c-fos in males only. Are you able to address which populations account for the increase in c-fos in predator behavior in males, and whether these cells could be the OTP cells?

The reviewers also raised an interesting question as to whether the OTP+ population is the missing population that responds to predator odor in males and whether we could conduct an electrophysiological analyses on these cells.

1) As to the best of our knowledge, there is no OTPcre or OTP reporter mouse line currently available which would have greatly facilitated our ability to identify and patch clamp these neurons as OTP+ cells in the adult MeA represents a very small population.

2) To address the behavioral question, we conducted new experiments to evaluate whether the OTP+ population was activated (c-fos+) during male and female predator exposure behavioral assays. We found that while there were significantly more OTP+/c-fos+ cells during male predator odor exposure as compared to control, when comparing the percent of male OTP+ cells that were activated (c-fos+) we did not observe significant differences between benign and predator avoidance conditions. However, when analyzing activation in female brains, we found an increase in the number of OTP+ cells activated as well as an increase in the percent of OTP+ cells activated vs benign. Therefore, OTP+ cells are being activated in the female MeA; nonetheless, the male predator odor activated population remains to be elucidated. This new behavioral data is shown in new Figure 7—figure supplement 1.

Reviewer 3 believes that it would be helpful to indicate the percentage of OPT+ and Dbx1+ cells, and the expected overlap by random chance, even though Foxp2+ and OPT+ do seem to be largely non-overlapping as they occupy virtually non-overlapping spatial locations.

Embryonically, OTP+ cells are segregated from the Dbx1-derived and Foxp2+ populations. This segregation generally persists into adulthood with overlap of the OTP+ population of approximately 3-5% of Dbx1-derived or Foxp2+ cells (new Figure 1—figure supplement 2). Therefore, as there was almost no-colocalization, we did not consider that there is any possibility of co-localization by random chance.

*Other comments that could be addressed textually:*

Reviewer 1 requests being mindful of the concept of sex differences (in c-fos expression), but not terming these sexual dimorphisms, even though reviewer 2 speaks of "demonstrate sexually dimorphic activation of specific lineage-derived populations in the medial amygdala".

This reviewer also called for citing a previous study on the dissonance between estrogen receptor and ligand, along with different expression of aromatase.

The reviewers thought that Figure 8 was too speculative and that Figure 8—figure supplement 1 be used instead.

We thank the reviewer for pointing out our misuse of the term sexual dimorphism. As suggested, we have replaced it with the term ‘sex-differences’. In addition, as suggested, we have now included a citation of a review of the studies of the Japanese quail brain (Balthazart et al., Front Endocrinology, 2011) and integrated a brief conceptual understanding of this topic into the Discussion section. Furthermore, as suggested, we have removed the overly speculative Figure 8 and replaced it with the previous supplementary summary schematic, which is also now updated to include the new OTP predator odor data.

Reviewer #3:

[…]

1) When understanding the overlap between two types of cells, the absolute number is not very informative. The percentage of cells of double labeled cells over single labeled cells provides some understanding of the overlap but it does not address the real question – are the two populations under investigation preferentially overlapped or not? For example, imagine that Dxb1 population accounts for 50% of the total number of cells and the Esr1 accounts for 50% of the total number of cells, if the double labeled cell is 25% of Dbx1 cells, then the Esr1+ cells are preferentially not derived from the Dbx1 cells. However, if Esr1 account for 10% of the total population, then the Esr1 is preferentially derived from the Dbx1 cells. Without knowing the percentage of the Esr1 cells, it is unclear what the 25% overlap actually means. Thus, throughout the paper, the authors should include information regarding the percentage of cells labeled by gene1, the percentage of cells labeled by gene 2 and the percentage of double labeled cells. Whether the two genes are preferentially labeled or not could be calculated as double%/(Gene1% xGene2%). A value over 1 means that the two genes are preferentially overlapped and a value below one will mean the opposite. With multiple animals, it can be calculated whether the preferential (or non-preferential) overlap is statistically significant or not.

We now provide new data showing the percent contribution of the Dbx1-derived and Foxp2+ cells to each sex steroid pathway marker (ERα, Aromatase, AR) (new Figure 4—figure supplement 2). We have also now conducted the analysis showing the percent contribution of OTP+ population to the percent total of the Dbx1-derived and Foxp2+ populations (revised Figure 1—figure supplement 1).

2) While the authors analyzed the sexually dimorphic involvement of the Dbx1 and Foxp2 cells in social behaviors, the basic characterization of the Dbx1 and Foxp2 cells in males and females is lacking. Does the percentage or the absolute number of Dbx1 or Foxp2 cells differ between sexes?

We have performed non-stereological analysis of the number of Dbx1-derived and Foxp2+ cells in the MePD and MePV and we did not observe any significant changes in the absolute number of cells between the populations. However, a conclusion as to whether these cells are indeed different between sexes would require a much more rigorous stereological analysis, therefore at this stage we would prefer not to comment in the manuscript on any potential differences, or lack of differences.

3) Regarding the overlapping of the Dbx1/Foxp2 with behavior induced Fos, there are several questions that could be readily addressed with the data. First, is the Dbx1 or Foxp2 cells preferentially/non-preferentially activated by the behavior (e.g. aggression)? For example, in Figure 5, the number of Dbx-1 cells expressing aggression induced Fos appears to be quite low. It is about 5% of the total Fos cells based on the graph. Although the Dbx1% is not reported, based on Figure 1, it appears to be quite dense, probably over 20% of the total cells. If so, the Dbx1 cells seem to be preferentially not activated during aggression. This should be analyzed and addressed. Second, to address whether the overlap between the Dbx1/Foxp2 and Fos is sexually dimorphic or not, it is better to compare the value of double%/(Gene1% xGene2%) rather than the double%. The double% is less informative as this value will be affected by the Fos% and the Dbx1% or Foxp2%. If Fos% is sexually dimorphic to begin with, it does not mean much if the double% differs between sexes.

We have now performed additional analysis by assessing the percent of Dbx1-derived or Foxp2+ cells that were activated (c-fos+) during aggressive, mating and predator avoidance behaviors and comparing to their respective controls. This new analyses is shown in revised Figure 5–Figure 7. This data is supportive of and strengthens our original findings with regard to numbers of cells activated in mating and aggression, but suggests a more complex interpretation with regard to predator odor, results of which are discussed in the revised Discussion.

4) Given the different number of Fos cells in Foxp2 and Dbx1 populations (e.g. Figure 5), it is likely that the distribution of Fos is biased topographically in the MeA. It will be helpful to present large view images of the Fos induction patterns. This will help the readers to understand potential reasons for overlap/non-overlap.

Images of the c-fos induction pattern are now present in new Figure 5—figure supplement 1.

5) The statistics in the paper should be double checked or the number of animals needs to be increased to provide sufficient statistical power. For example, in Figure 4, there is a strong trend (if not yet significant) that the co-labeled% in female Dbx-1 population is higher than the rest of the groups. If one perform a t-test between the male and female Esr1 and Dbx1 double labeled%, it should be significant given that there is no overlap in values and the variability is not huge.

We previously used ANOVA for the statistical analysis of Figure 4 to account for multiple group comparisons. As per the reviewer’s suggestions, we have performed t-tests instead and changes are now reflected in revised Figure 4 and changes made to the Materials and methods to reflect this revised analysis.

6) Lastly, the authors should provide the Figure number in each figure file to facilitate the review process.

The figure numbers have been included in every figure.

Overall, this is a good study with interesting findings. However, the results can be better analyzed to better understand the potential molecular features and behavioral relevance of Dbx1 and FoxP MeA cells.

[Editors' note: further revisions were requested prior to acceptance, as described below.]

You have done an excellent job of responding to the reviews of your revised manuscript with added additional data and new analyses. The study indeed contributes to an understanding of the developmental logic for processing of innate behaviors by different MeA neuronal subtypes. However, there are some remaining issues that need to be addressed before acceptance, as outlined below:

1) You changed the statistical tests in new Figure 4:

You state in response to Reviewer #3 Critique #5, that some groups in Figure 4 looked significantly different, but no such difference was detected by ANOVA, and then you proceeded to conduct 2-tailed t-tests. The reviewers believe that there is no justification for this approach, as one of the recurrent themes of the manuscript is the presence and absence of sex differences. Neither can be concluded when males and females are analyzed separately, only within sex comparisons can be made. In Figure 4 the bars indicate a significant difference between males and females but this panel alone involves at least 3 t-tests, which is an inappropriate use of multiple comparisons without correction. This is true for most of the data in this figure, and perhaps others. The appropriate analysis is an ANOVA with sex as a factor and only when there is either a main effect of sex or a sex by other factor interaction can post-hoc tests be done.

Further, for use of one way ANOVA, the data need to be normally distributed and exclude outliers. One outlier can throw off the statistics and provide a false negative. While pairwise comparison should not be done without finding significance in ANOVA, it is essential to check whether ANOVA is correctly used here.

Given the very small n's in some of these endpoints, which is understandable given the labor intensiveness of the co-localization analyses, it is imperative that correct statistics are conducted. In this instance it is probably better to miss a sex difference or effect than it is to incorrectly claim one based on a small n and inappropriate statistics. We thus encourage you to seek advice from someone well-versed in statistical analysis to help you work through this, report back to us, and revise the manuscript accordingly.

We have consulted with our in-house director of the DC-IDDRC Biostatistics Core. In our reanalysis, we now first determined whether the data was normally distributed by performing the Shapiro-Wilk test for normality and we also excluded outliers (any data points two standard deviations away from the mean). For Figure 4, we now performed a two-way ANOVA with interaction for sex and neuronal subpopulation for data that was normally distributed. If there was a main effect or an interaction for sex, then we performed multiple comparisons. If data was not normally distributed, we performed a non-parametric test, in this case a quantile regression analysis (Figure 4). For Figure 4—figure supplement 2, we also checked for normal distribution and performed a two-way ANOVA for normally distributed data (Figure 4—figure supplement 2) or a quantile regression analysis for non-normally distributed data (Figure 4—figure supplement 2). A detailed description of the models used for the data analysis of Figure 4 and Figure 4—figure supplement 2 can be found in the Materials and methods section. In addition, for Figure 5, Figure 6 and Figure 7 we also performed the Shapiro-Wilk test to determine normality. For data normally distributed we performed an unpaired t-test or a one-way ANOVA followed by Tukey’s post-hoc and for data that was not normally distributed we performed either the Mann-Whitney test (data with two groups) or the Krustal-Wallis analysis followed by Dunn’s corrections (data with 3 groups). Details of the new statistical analyses are described in the respective revised Figure legends and the Materials and methods sections.

This new analyses revealed the following differences from our previous analyses:

1) Figure 4: We now find that there are no sex-specific differences in the total *Dbx1-*derived subpopulation expressing estrogen receptor α. Also, we do now find sex-specific differences in the expression of androgen receptor in the *Dbx1-*derived subpopulation.

2) Figure 4—figure supplement 2: Related, we now find sex-specific differences in the co-expression of *Dbx1-*derived cells with androgen receptor, over the total androgen receptor subpopulation, consistent with the above finding in Figure 4.

3) Figure 5: no statistically significant difference is now found between one of the controls (“no pups” condition) and maternal aggression (in the previous version this was significant). Importantly, statistically significant differences remain between all naïve controls and experimental conditions in both sexes.

4) Figure 7: no statistically significant difference is now found between the number of cfos+;*Dbx1*- derived cells during predator odor exposure compared to benign control. In our previous and current analyses we did not find any significant difference between the percentage of cfos+;*Dbx1*-derived cells/cfos+ cells (Figure 7). Thus, our previous conclusion that neither the *Dbx1*-derived nor Foxp2+ populations were activated during predator odor remains unchanged.

Importantly these newfound differences do not affect our conclusions in any manner.

2) Also in relation to the graphical presentation of statistics: in Figure 5, and others, the use of bars above the histograms is confusing as usually an open-ended bar indicates all groups below it are different from each other. There are clearer ways to indicate post-hoc significant differences between 2 groups, but hard to explain in writing: either a combination of letters, or bars with end feet that point to only the 2 groups. You are encouraged to look at other papers with ANOVA's for some ideas.

We included in all figures that contained three groups bars with end feet to point out clearly which groups showed significance post-hoc.